

# A method to employ the spatial organisation of catchments into semi-distributed rainfall-runoff models

Henning Oppel[1], Andreas Schumann[1]

[1]Institute of Hydrology, Water Resources Management and Environmental Engineering, Ruhr-University Bochum, Bochum,
44801, Germany

*Correspondence to*: Henning Oppel (henning.oppel@rub.de)

**Abstract.**

A distributed or semi-distributed deterministic hydrological model should consider the hydrological most relevant catchment characteristics. These are heterogeneously distributed within a watershed but often interrelated and subject of a certain spatial organisation which results in archetypes of combined characteristics. In order to reproduce the natural rainfall-runoff response the reduction of variance of catchment properties as well as the incorporation of the spatial organisation of the catchment is desirable. In this study the with-function approach is used as a basic characteristic to analyse succession of catchments characteristics. With this technique we are able assess the context of catchment properties like soil or topology along the stream flow length and the network geomorphology, giving indications on the spatial organisation of a catchment. Moreover, these information and technique has been implemented in an algorithm for automated sub-basin ascertainment, which includes the definition of zones within the newly defined sub-basins. Aiming to provide sub-basins that are less heterogeneous than common separation schemes. The algorithm is applied on two parameters characterising topology and soil of four mid-European watersheds. Resulting partitions indicate a wide range of applicability for the method and the algorithm. Additionally the intersection of derived zones for different catchment characteristics could give insights on sub-basin similarities. Finally a HBV$_{96}$-case study also proved the benefits for modelling of the new subdivision technique.

## 1 Introduction

Hydrological models are instruments for structuring the knowledge about hydrological processes in their dependencies from watershed characteristics. For the set-up of these models several initial decisions have to be made: what type of model has to be used, what temporal resolution would be appropriated, which spatial resolution of the model is necessary and useful and how can the model be parametrised? It is clear that all of these choices affect the effort of the model and all choices have to consider the modelling purpose. Moreover, these choices are interrelated, for example predominant soil properties define the dominant runoff process and should, hence, define the used model. Therefore conceptual models of natural watershed require its subdivision into spatial units as homogenous as possible. Hydrological modelling is the attempt to specify hydrological processes quantitatively under consideration of boundary conditions. These boundary conditions are mainly determined by





spatial heterogeneous distributed catchment characteristics. There are several approaches to consider this heterogeneity in models and to specify more or less homogenous units.

One option is a subdivision of a river basin into sub-basins which has to be modelled separately. The common approach for such a subdivision is usually based on the available hydro-meteorological data, thus the correct criteria would be the spatial
heterogeneity of hydrological characteristics within the river basin. If the heterogeneity is low, neighbouring basins can be modelled together. If sub-basins are very different from another it should be described separately with a specially adapted model which considers its specific characteristics (e.g. an urban watershed model). Afterwards each sub-basin is handled as a unique modelling instance that should comprise a minimum of heterogeneity (in sense of the key catchment characteristic). In this way each sub-basin has its own model and/or parameters to adjust the models to mimic its natural response.
Another way to consider spatial heterogeneity within watershed is to subdivide the catchment into so-called hydrologic response units (HRUs). A single unit merges areas, or cells, within a basin comprising similar characteristics independent from their respective spatial allocation, each unit is a unique modelling instance. The crucial assumption for each HRU approach is that the variation of the hydrological process dynamics within the HRU must be small compared with the dynamics in a different HRU (Flügel, 1995). HRUs are formed by overlaying coverages which specify areas of the watershed which fulfil
common physiographic criteria. The delineation of HRUs by combinations of these coverages demands a subdivision of characteristics (soils, land-use and vegetation types, topography, and geology) into classes to keep the number of HRUs manageable. The selection of criteria, their subdivision into classes and the degree of heterogeneity which is accepted within these classes depend of on the insight gained from the hydrological systems analysis. Based on the chosen technique, or purpose of HRUs they omit the actual spatial allocation (Lindström et al., 1997; Schumann et al., 2000) or define coherent units (Dunn
and Lilly, 2001; Soulsby et al., 2006; Müller et al., 2009; Nobre et al., 2011; Gharari et al., 2011). However, some of these models try transfer geological information (Müller et al., 2009; Soulsby et al., 2006) or topographical information (Nobre et al., 2011; Gharari et al., 2011) to hydrological processes and assume homogenous conditions of remaining parameters.

Another option to handle spatial heterogeneity in hydrological modelling consists in the use of a distributed catchment characteristic as a covariant measure which supports the spatial distribution of a lumped state variable. An example is the use
of the topographic index in the well-known TOP-model as a characteristic of the spatial variability of the soil water content (Beven et al., 1984).

Since GIS layers are widely available it is a steady tendency to incorporate this data into the ascertainment of spatial units. Most approaches are based on topography (Band, 1986; Moore and Grayson, 1991; Vogt et al., 2003; Lai et al., 2016) and focus on the extraction of stream networks and the network connectivity. The focus of these processes is caused by widely
employed topology driven modelling concepts (Beven and Kirkby, 1979; Rodríguez-Iturbe and Valdés, 1979). Especially the development of the geomorphologic instantaneous unit hydrograph (GIUH) as well as its enhancements like the geomorphological dispersion (Rinaldo et al., 1991; Gupta and Mesa, 1988) required sophisticated stream network derivation and analysis. Methods introduced by Band (1986) or Verdin and Verdin (1999) accounted for this requirements. Moreover Snell and Sivapalan (1994) used the width-function, introduced by Kirkby (1976) to describe the geomorphic structure of





networks. They showed that GIUHs based on the width function represented the geomorphic dispersion better than GUIH derived from Horton laws (Robinson et al., 1995; D'Odorico and Rigon, 2003; Rigon et al., 2016). This made a further step to incorporate remote sensed data in describing the organisation of catchments. All beforehand described methods are based on gridded digital elevation models (DEM). Other methods try to identify streamlines derived from DEM-shapes, representing

contour-lines (Moore and Grayson, 1991; Lai et al., 2016) that are used subsequently as modelling instances.

A combination of the different methods to consider spatial heterogeneity of watershed characteristics is proposed in the following. It is based on the on a consideration of pattern resulting from the spatial organisation of catchments. Sivapalan (2005) pointed out that the organisation of a catchment has a fundamental influence on the hydrologic system. He defined the organisation of a catchment as patterns of symmetry between soil, topography and the stream network. These patterns could

give insight to underlying mechanisms that induce discharge behaviour. Especially the connection of soil data with the flow path lengths at hillslopes could bring a deeper insight into processes of lateral flow distribution (Grayson and Blöschl, 2001). In this study we will present a method to address these pattern by a combination of the width-function (Kirkby, 1976; Mesa and Mifflin, 1986) with soil properties like pore volume and topographic characteristics like surface slope. In contrast to classic methods for spatial pattern evaluation (like Point-by-Point or optimal logical alignment methods (Grayson and Blöschl, 2001))

we retain the allocation of catchment properties. This analysis reveals the organisation of the watershed and gives indications of gradations of spatial heterogeneity which are useable for the set-up of an appropriated model structure.

Based on this method we developed an algorithm for automated sub-basin ascertainment. Our aim is to incorporate the spatial organisation of watersheds into the spatial structuring of a semi-distributed model and to assess its benefits for model performance. The aim of the proposed algorithm is set to offer a basin partition with a minimum of heterogeneity by a minimum

of sub-divisions, i.e. to reduce the computational effort in cases of hydrological modelling.

The proposed algorithm has been applied in a case study to four meso-scale mid-European watersheds and in a $HBV_{96}$ modelling application in one of these basins. Remaining heterogeneity and the modelling performance of the proposed subdivision scheme were compared to a common subdivision / modelling setup.

This paper is organised in 4 parts. First we introduce the data (Sec. 2), because we give some references to observations in

some of the basins during the description of the methodological development. This is in fact the second part (Sec. 3). It involves observations, considerations and techniques to assess spatial patterns of catchment characteristics and their spatial organisation. Here the sequence of the proposed algorithms is presented which was used to incorporate the spatial organisations into the model. In the third part (Sec. 4) we will analyse our tools, i.e. check its applicability and its limits. The last part (Sec. 5) describes the application of the method, comprising subdivision case study and modelling application.

**2 Data**

The proposed methods are results of GIS-based catchment analysis, hence, we need to introduce our data base first. Generally four catchments were chosen to develop and test our methodological approach, while one catchment served as a development





catchment and the remaining for validation. Our development catchment is the basin of the Mulde River (Fig. 1, left). It is located mostly in eastern Germany and with a small part in north Czech. In its southern part it comprises the mid-range mountainous region of the Ore Mountains. With a size of 6170 km² it is the largest catchment used in this study. The three other catchments are the catchments of the rivers Regen (A = 2613 km², Fig. 1, lower mid), upper Main (A = 4224 km2, Fig. 1, upper mid) and the Salzach (A = 5995 km², Fig 1, right). While the Mulde, Main and Regen basins are mid-mountainous catchments the Salzach basin is an alpine catchment. All catchment have different geomorphologic structures and river network types. While in the first three catchments, higher mountains are nearly exclusively located at the outer catchment boundaries, the Salzach catchments contains three high mountains located at the centre of the catchment. The two main tributaries encompass these mountains. While the basin of the Mulde River has a nearly continuous increase of slope and elevation from north to south, the topography of the remaining catchments is much more heterogeneous.

For the proposed methods and algorithm, at least a digital elevation model (DEM) is essential. For this study we used a gridded DEM derived from the Shuttle Radar Topography Mission (SRTM) with a regular resolution of 100 meters. By application of the D8-algorithm the required data like flow directions, flow length and flow accumulation (i.e. number of cells draining to the respective cell) were calculated (Jenson and Domingue, 1988). For the catchment of the Mulde River a proved digital river network was available. Stream networks of the remaining basins were calculated via flow accumulation algorithms. To characterise the soil characteristics of the German catchments, a gridded soil data map from the German Federal Institute for Geosciences and Natural Resources (BÜK200) and CORINE land coverage data (CLC) (Bossard et al., 2000) were used. Pedo-transfer functions (Sponagel, 2005) were applied to transfer these information into gridded data about (available) water capacities (AWC) and other soil storage parameters as well as hydraulic conductivities. In case of the Salzach basin precast pore volume data provided for Europe along with the LARSIM-ME model were used, due to a lack of soil data (Bremicker, 2016). The used soil and topography input data comprises a certain amount of uncertainty because they are derived data. The uncertainty of the input data is neglected in the performed case studies. Pore volume data is depicted in Fig. 2.

## 3 Methodological development

In this section an algorithm that characterises the heterogeneity of regions and applies techniques to efficiently subdivide the watershed is introduced. Since the sequence of the algorithm is better intelligible if the used techniques are known we first introduce all required new techniques and methods and then present the algorithm.

First, the underlying approach of the distance-factor function for the assessment of spatial organisations (Sec. 3.1) and subsequently the tools of the algorithm (Sec 3.2) are introduced. The concluding Sec. 3.3 presents the sequence of the proposed algorithm. All tools are only briefly introduced, a detailed description of the applied methodologies can be found in the Appendix.





### 3.1 Distance-factor function

Throughout the development and ongoing research concerning the geomorphologic instantaneous unit hydrograph (Rodríguez-Iturbe and Valdés, 1979; Gupta et al., 1980) the width-function as introduced by (Kirkby, 1976) and its further development the area- or weight-function (Mesa and Mifflin, 1986; Snell and Sivapalan, 1994; Robinson et al., 1995) has been used to

describe the distribution of runoff producing area with respect to flow distance from the outlet (Mesa and Mifflin, 1986). The weight-function, hence, gives the probability distribution for a uniform areal precipitation intensity for the choice of a flow path (Snell and Sivapalan, 1994). Since we know the flow path and distance we are able to describe the hydrograph at the outlet of a basin under the assumption of a uniform velocity.

But velocity is not uniform in a basin. It is dependent on its surrounding medium (soil, air, other water particles) and the

medium condition (dry or wet/empty or full) and a wide range of other impact factors. Therefore, let us describe the transformation from the arrival of precipitation at its flow path to the outlet of the catchment as trail-function. It merges losses and retention of water along the flow path. The detailed description of the trail-function is topic to a hydrological model and, at this point, bound to the deliberate choice of the user.

However, nearly all hydrological models require information about a catchment characteristic(s) or at least homogeneous

conditions of a single characteristic (in most cases soil properties). Coming back to the idea of the weight-function, it seems worthwhile to develop a method to assess an arbitrary catchment characteristic in the same manner.

We propose the *distance-factor function* to analyse the distribution of an arbitrary catchment property $C$ with the flow path. For this purpose we have to split the flow path into the part of a path through (or along) the hillslope (hillslope flow length $x_H$) and the part within a stream till the outlet (stream flow length $x_S$). Note that stream cells have a hillslope flow length of 0 and

hillslope cells comprise the $x_S$ of their draining stream cell. Further remind that the term flow length refers to the actual length of the path water has to travel to the outlet, its calculation is based on the D8-algorithm (Jenson and Domingue, 1988).

To assess groups of hillslopes and to account for the non-continuousness of grid-based distances we substituted the estimated flow lengths ($x_S$ and $x_H$) by distance classes. The stream flow length is subdivided into multiples of the length $\Delta s$ (for $x_S$) and the hillslope flow length into multiples of the distances $\Delta o$ (for $x_H$). In this way defined distance classes split the basin into

stripes.

Within each distance-class multiple values of the considered catchment property are present. Within these classes we are now able to show an expected value $E(C)$ and/or the standard deviation $\sigma(C)$ of the respective characteristic:

$$E(C)_i = \frac{1}{w(i \cdot \Delta s)} \sum_{x=i \cdot \Delta s}^{(i+1) \cdot \Delta s} C_x \qquad (1) \quad \text{and} \qquad \sigma(C)_i = \sqrt{\frac{1}{w(i \cdot \Delta s)} \sum_{x=i \cdot \Delta s}^{(i+1) \cdot \Delta s} \left( C_x - E(C)_x \right)^2} \qquad (2)$$

Where $i$ indicates the number of the distance class that is multiplied with the width of the distance class $\Delta s$ and $w(i \cdot \Delta s)$ is the

number of values (or grid cells) within the class. Please note that w(x) is the non-normalised value of the area-function (Snell and Sivapalan, 1994). Figure 3 shows the application of both values (Eq. 1) for AWC in the Mulde catchment. Expected values and a 1-$\sigma$-range are plotted against the stream flow distance to the outlet. Distance classes with lower and higher variance are





clearly visible in this figure as well as the succession of lower values to higher expected AWC values with increasing stream flow lengths.

With this approach we are able to assess the arrangement of catchment properties with the flow-path which will be, in case of a non-random arrangement, be referred as the spatial organisation of the basin. Please note that we will always co-notate which
type of distance-factor function (expected value or standard deviation) has been applied.

## 3.2 Tools for automated sub-basin ascertainment

The example of a distance-factor function in Fig. 3 demonstrates that in some distance classes the AWC-values are similar as the standard deviation is low. This can be caused as the area of the distance class is small (which is the case close to the outlet and in the furthest distance class) but also as the class belongs to the same region (low lands, mountainous regions). In the
example the distance classes between 50 and 170 km are more heterogeneous which becomes evident from the higher standard deviations. In order to minimise heterogeneity in these regions, a further subdivision is needed in this parts of the catchment.

For the ascertainment of more homogenous sub-basins an algorithm has been developed which is based on the following components:

1. An objective function which identifies the needs and (if necessary) the region of further subdivisions.
2. A tool to specify the subdivision points in the selected region for subdivision of the catchment.
3. An evaluation strategy to assess performed subdivisions.

The following subsections give a brief introduction to these three functionalities, before the sequence of the algorithm is described. More details about the introduced tools are given in Appendix A.

### 3.2.1 Objective Function

The standard deviation $\sigma(C)$ is used an indicator for the heterogeneity of the sub-areas. From the result of the distance-factor function for $\sigma(AWC)$ of the Mulde (Fig. 4) the $1 \cdot \sigma(C)$-range clearly indicates regions comprising higher heterogeneity. These regions should be considered for subdivision. To make this tangible for the GIS-based algorithm, a threshold value $\Omega$ has been introduced that states whether a distance class is to categorise by (relatively) "low" or "high" standard deviations. The threshold $\Omega$ is derived from the data by:

$$\Omega = \frac{\sum_{j=1}^{N_S} \omega_j^{\,e} \cdot \sigma(C)_j}{\sum_{j=1}^{N_S} \omega_j} \qquad (3)$$

Where $N_S$ is the number of distance classes in the basin, the exponent $e$ is the parameter to a (non-)linearity factor and $\omega$ a weighting factor is defined as:



$$\omega_j = \frac{\sigma(C)_j - \max_{N_S}\left(\sigma(C)\right)}{\min_{N_S}\left(\sigma(C)\right) - \max_{N_S}\left(\sigma(C)\right)} \tag{4}$$

The actual objective function $Z$ is the number of distance classes that comprise a standard deviation higher than the threshold $\Omega$, which can be written as the cardinality of the set of classes fulfilling this condition, which the algorithm will try to minimise:

$$Z = \left\| [0,...,i,...,N_S \mid \sigma(C)_i > \Omega] \right\| \rightarrow \min \tag{5}$$

The proposed algorithm tries to minimize this value. Coherent distance-classes above the threshold are considered as *regions* of high variance of the characteristic of interest that require a further spatial subdivision. If the standard deviation in coherent distance-classes stays below the threshold, these regions will be marked as low variance regions.

As index $N_S$ in Eq. (3) indicates, threshold $\Omega$ is calculated for the entire catchment and does not necessarily represent the true value of $\sigma(C)$ for homogeneous sub-basins. It only delimits between regions of high and low variances. Dependent on the

purpose it is recommended to apply the proposed algorithm multiply in the process of subdivision for a stepwise reduction of $\Omega$ and hence the remaining variance.

Note that each application will lead to a higher number of subdivisions.

### 3.2.2 Subdivision tools

To explain the proposed tools for subdivisions in Fig. 5 a synthetic basin with its stream network, distance-classes and an

arbitrary characteristic is shown. This basin will be used here (and in Appendix A) for a visualisation of the proposed tools.

The application of the objective function has three possible outcomes:

1. The standard deviations in all distance-classes stays below threshold $\Omega$,
2. Only parts of the classes have values of $\sigma(C)$ smaller than $\Omega$,
3. Standard deviations of all classes are greater than $\Omega$.

The first case indicates that no further subdivision is required. The second case indicates that a parts of the flow path comprise nearly homogenous characteristics. This case is depicted in Fig. 6a. Hatched cells indicate the low variance region. Since this region is homogenous it will not require any further handling and can be separated from the rest of the basin, which is more heterogeneous.

For this purpose a tool called *Detachment* has been developed to clip low from high variance regions. The tool locates the ideal

drainage points whose watershed covers a maximum of the low variance regions. The search for these separation points is made iteratively close to the transition from low variance to high variance region. In the Fig. 6b the result of *Detachment* is shown for the introduced synthetic catchment. Black dots indicate the identified drainage points, hollow points indicate potential separation points which were rejected in the course of iterations. After applying the *Detachment* tool, three sub-basins are defined. One containing the low variance regions that does not require any further treatment and the other two sub-basins

comprising the remaining parts of the catchment.





Both remaining heterogeneous sub-basins, in this example, consists of distance classes with standard deviations above the threshold. This can be caused by different spatial patterns in the sub-areas. On the one hand, parallel streams, or more specific neighbouring valleys with different vegetation, slope etc. can cause higher variance. On the other hand a zoning of hillslopes and higher elevated parts of the basin, framing the drainage network (e.g. gley horizons close to stream) results in higher values

of the standard deviations. Such patterns are a result of the co-evolutional formation of catchments (Blöschl et al., 2013; Sivapalan, 2005).

With regard to these two different reasons of heterogeneity, two different tools were developed. The first tool provides a subdivision at stream branches (because our perspective of analyses is directed upstream, downstream would be confluences) which defines new sub-basins at branching points of higher order streams. The *pruning* tool identifies branches by means of

the distance-factor function of flow accumulation (see Appendix A).

The second tool is a *zonal classification* scheme, designed to account for heterogeneity within a distance class that is not caused by neighbouring valleys. We provide three zone types whose individual extent is ascertained iteratively with the aid of three variables: Strahler order, distance to stream (compare $x_H$ Sec. 3.1) and heights. First zone type "*close to stream*" is determined by Strahler order and is intended to encase the stream-network. Which streams (beginning at the highest Strahler order and

adding lower orders successively) and to which extent (distance to the stream, beginning with $1 \cdot \Delta o$) close to stream zones are build is, as already mentioned, done iteratively. Second zone type, referred as "*transition*" zones, are on first instance defined as all cells of the sub-basins which are not defined as "*close to stream*". Subsequently all cells are reassigned by their height-value either into "*high elevation*"- (heights above the iterated threshold) or into "*transition*" (heights below threshold)-zones. Beforehand mentioned height threshold is iterated as quantile of the histogram of height of all non-"*close to stream*" zones.

In Fig. 6d & e the results of an application of both tools for the exemplary synthetic catchment are shown. Hatched cells in the lower left depiction are high variance regions that require partition. The *pruning* results in two additional drainage points and differentiates this region into three sub-basins. The *zonal classification* gives no new drainage points but specifies three zones belonging to the previously defined drainage points. Note that the algorithm has the opportunity the reject a zonal classification if the resulting variance is equal or higher than in the un-classed sub-basin. In this case all cells will be marked as "*None*"

zones.

Introduced tool-names will be used in Sec. 3.2 (and Fig. 7) were a detailed description and explanation of the algorithms sequence is given.

### 3.2.3 Evaluation scheme

After application of one of the previously introduced tools it has to be evaluated if its target has been achieved. If we recall

our target, minimise heterogeneity through the introduction of sub-basin and zones. As our assessment of heterogeneity is based on the evaluation of distance-classes, we can reformulate our objective as the minimisation of standard deviation within each distance class. No matter which tool has been applied, afterwards in some or all distance classes of the original sub-basin U multiple sub-basin / zones are present. Since standard deviation is calculated for each partition unit (sub-basin or zone)





individually, we are able to calculate the average standard deviation of neighbouring units within a distance class. This evaluation technique can be expressed as follows: Let $B$ be the number the sub-basins defined within the original catchment U. In the first step we estimate the standard deviation $\sigma(C)$ (Eq. 2) within each sub-basin within distance classes based on flow length to the outlet of this sub-basin.

In the second step the calculated $\sigma(C)$-values are transferred to the flow-length axis of the original watershed U. This is done for all sub-basins by adding the stream flow length between their points of confluence and the outlet of the basin. Finally the new standard deviation $\sigma_S(C)$ for the separated basin is calculated for each distance class of U as average of $\sigma(C)$-values assigned to the class of the initial sub-basin (U).

$$\sigma_S(C)_i = \frac{1}{\sum\limits_{j=1}^{B_i} w(i \cdot \Delta s)_j} \cdot \sum_{j=1}^{B_i} \sigma_j \cdot w(i \cdot \Delta s)_j \qquad (6)$$

Where $B_i$ is the number of distance classes and $w(i \cdot \Delta s)$ the number of cells within the distance-class $i$, which is always lower or equal to $B$. In comparison to the standard deviation of the unseparated basin $\sigma_U(C)$ the success of the partition can be measured, e.g. as the quotient of these values. With the new value $\sigma_S(C)$ the effort of each subdivision can be measured independent from the objective function (and its threshold).

### 3.3 Sequence of the algorithm

As the tools presented above are at this point incoherent. Their sequential application is described in this section. Therefore the sequence of the ACS-algorithm (**A**scertainment by **C**atchment **S**tructure) will be explained step-by-step following the sequence shown in the flowchart in Fig. 7. Before this is done our considerations leading to the present sequence are specified as follows:

1. Homogeneous regions should be separated by the algorithm from the remaining basin.
2. It is desirable to subdivide a basin at major branches/confluences of the river network into sub-basins.
3. For high variance regions the two options *pruning* and *zonal classification* should be carried out in competition, since the origin of high variance is unknown.
4. The results of both technique (*pruning* or *zonal classification*) will be compared. The subdivision yielding a lower $\sigma_S(C)$ will be saved, other results will be discarded.
5. *Zonal classification* is additionally applied in cases where only low variance regions are present. Thereby, independent from the need of $\sigma(C)$ reduction, all ascertained sub-basins will comprise a zonal classification.

At the very beginning of the sequence, on initialisation of the algorithm, we consider just one drainage point at the outlet of the basin. After calculation of its watershed and the determination of the width-function of accumulated partial areas it will be evaluated if we the need to consider a major branch (See Appendix for detailed description of this procedure). Major branches
are intended to account for larger, main rivers within a catchment. Since larger rivers comprise a larger number of cells draining into them, branches/confluences of rivers are easily to differentiate from smaller branches. If the test for major branch is true,





the tool *pruning* will be called to specify two new drainage points (sub-basin outlets). They are saved and put on the schedule of the algorithm for later examination.

If a subdivision has been performed the algorithm starts again at the previously used point, but now it will only look at the watershed between the actual and the newly defined drainage points. If no further major branch is present, the algorithm

calculates the standard deviations of the characteristic of interest and the objective function, to estimate the number of distance-classes above the threshold $\Omega$.

There are three possible outcomes (as one might recall from Sec. 3.2.2): Standard deviation in none, some, or all classes is above the threshold. For each outcome the algorithm has an option:

- If no class is above $\Omega$, no partition is required and hence the analysis of this part of the basin is completed. But to

10          fulfil consideration 5, the *zonal classification* is started and their results are saved. The algorithm proceeds and examines the next drainage point on its schedule.

- If only a couple of classes are above $\Omega$, a partition is required but there are other areas that do not require subdivision. These areas are clipped by means of the *detachment* tool. One or more new drainage points are defined and added to the schedule. The algorithm starts again at its active point.

- In the last case operations as defined in point 3 & 4 of our consideration will be started. Both tools try to lower the heterogeneity trying to handle different roots of variety. Result giving a lower remaining variance will be saved, the other result will be discarded. The algorithm starts again at its active point.

This procedure is repeated until all drainage points have been examined by the algorithm. The concept of the schedule and that each basin, or sub-basin, is analysed again after each subdivision, gives the option to apply a pre-partition. This might be

applicable if an existing structure (like a gauging network) is analysed for further improvements or just for zonal classification.

# 4 Method analysis

In this section we will analyse the outcome of application of the proposed algorithm on our case study catchments. First we will take a qualitative look at the ascertained sub-basins and zones to assess similarities between catchments and characteristics. Afterwards we will have a quantitative look at the performance of the algorithm in sense of its objective function.

The ACS-algorithm has been applied to all four catchments for pore volume and surface slope. Ascertained sub-basins and zones are shown in Fig. 8 a-d (Mulde and Regen) and Fig. 9 a-d (Main and Salzach). Additionally the distance-factor functions for standard deviation $\sigma(C)$ for the respective characteristic $C$, prior and after partition of the catchment, are shown.

## 4.1 Qualitative Evaluation

On first sight it is visible that the proposed zonal classification showed applicability to all catchments. Only one sub-basin in

the Mulde catchment rejected a zonal classification (Fig. 8a, red sub-basin). Additionally the defined sub-basins for both characteristics (same catchment) are comparable and often identical. This is mostly caused by the subdivision at major branch.





Nevertheless, differences in the number and extent of defined sub-basins are visible. But more striking are the similarities and dissimilarities of the defined zones.

Both applications within the Regen (Fig. 8 c, d) and Main (Fig.9 a, b) catchment resulted in similar patterns of "*high elevation*" zones. In the Main catchment even the extent of "*close to stream*" zones (CTS-zones) is very similar. Wide-spread CTS-zones

seem not to be necessary for pore volume in the Regen catchment. If we have a closer look at the CTS-zones for pore volume in all catchments and compare them to the maps of pore volume (Fig. 2) we can identify a pattern. Especially in the Mulde and Salzach catchment very narrow CTS-zones have been defined most parts of the basin. The occurrence of these narrow zones coincides with the presence of belts of different pore volumes around a major streams (best visible in the Mulde catchment, Fig. 2, middle region). The soil patterns are not visible in the Regen and play a minor role in the Main catchment, hence extent

of CTS-zones is more likely to be extensive.

The same analysis for surface slope zones shows that these zones are, for the most cases, more extensive than the pore-volume zones and follow the valley structure of the DEM (Fig. 1). CTS-zones cover the streams and floodplains at the bottom the valleys, "*transition*" zones cover the hillslopes and "*high elevation*" zones cover higher located plains.

The interplay of zonal extent is best visible in the Salzach catchment which is in all of its characteristics the most diverse (very

high mountains with high slopes, soils with high and nearly no storage capacity). A comparison of the outcome for pore volume and slope (Fig 9 c, d) with the respective maps (Fig. 1 & 2) shows that the "*high elevation*" zones for pore volume cover the bare soil and rock formations with nearly no pore volume and in case of slope application the higher elevated parts of the mountains with only little slope. "*Transition*" zones cover in this case the steep hillslopes between the (comparably) flat valley bottoms and high plateaus. In case of pore volume they capture the mid-range soils between the mountain top and the

floodplains which are in both applications encompassed by CTS-zones.

From this analysis of spatial natural patterns and patterns in the ascertained sub-basins and zones we can draw the conclusion that the outcome of the ACS-algorithm is bound the spatial organisation of the considered catchment.

## 4.2 Quantitative Evaluation

We might have shown that the proposed algorithm can mirror the spatial organisation of the catchment, but we have not

evaluated if the heterogeneity of the specific characteristic has been lowered. As stated in Sec. 2.4 the objective function of the algorithm is the reduction of the number of distance-classes comprising a standard deviation $\sigma(C)$ above the threshold $\Omega$ (of the characteristic $C$). To evaluate the fulfilment of this target we address the reduction of $\sigma(C)$, defined as the difference of summed standard deviation of the unseparated catchment $\sigma_U(C)$ and the separated basin $\sigma_S(C)$:

$$\alpha_1 = \frac{\sum_{i=0}^{N_S} \sigma_{U;i}(C) - \sigma_{S;i}(C)}{\sum_{i=0}^{N_S} \sigma_{U;i}(C)} \tag{7}$$



The total reduction $\alpha_1$ is calculated over all distance classes $Ns$ of the original unseparated catchment and is normalised by the sum of $\sigma_U(C)$. Standard deviation of the separated catchment $\sigma_S(C)$ is calculated according to Eq. 6. Beside this evaluation we propose a second performance index to evaluate if our target has been reached just scantly or more solid. Second measure $\alpha_2$ is intended to show cases where the total heterogeneity has been lowered significantly, but still remains above the objective:

$$\alpha_2 = \frac{\sum_{i \in M(S)} \Omega - \sigma_{S;i}(C)}{\sum_{j \in M(U)} \Omega - \sigma_{U;j}(C)} \tag{8}$$

In Eq. 8 $M$ is the set of distance classes comprising a higher standard deviation than the threshold $\Omega$:

$$M = [0, ..., i, ..., N_S \mid \sigma_i \geq \Omega] \tag{9}$$

Performance criteria for all applications are tabulated in Tab. 1, additionally the number of ascertained sub-basins are shown (in terms of modelling/calibration effort relevant).

As already could be anticipated from the distance-factor functions of $\sigma(C)$ in Fig. 8 and 9 the performance of pore volume applications is in all cases (and both evaluation subjects) superior to slope applications. Total reduction is 2 to 6 times higher and ranges for pore volume from 46 to 65% total reduction of variance. The remaining variance above the threshold ranges from 0.9% to 25% and is superior, too. Difference lies between 8% (Regen) and 67% (Mulde). Although the reduction of variance for slope might be inferior, yet up to 25% of variance could be compensated by the ACS-partition of the basin.

If we focus on the cases with negative outcome we are able to identify some limitations for the algorithm. At first we will have a look at the slope application. Reduction is generally low and the remaining variance is high, but especially the outcome of the Mulde basin is inferior to all other (slope) applications.

The cause of this inferior performance might be the shape and arrangement of the catchment itself. In contrast to the other basins, the Mulde basin can be idealised as triangular. Several streams arise from the south of the basin and converge gradually heading to the north. Yielding nearly parallel situated sub-basins with the same spatial organisation of heights and slope. As it can be seen in the distance-factor function, the variance increases in upstream direction nearly continuously, equally distributed and remains on a (comparably) low level (see ordinate-axis of distance-factor functions for remaining basins in Fig. 8 and 9). Contrary to that, the remaining catchments offer different spatial patterns. Where headwater catchments with higher elevations and slope lie within same distance classes with plain catchment parts, offering a higher variance and, hence, a better opportunity

for subdivision.

Another inferior case is the pore volume application in the Salzach basin. Shape of the basin as well as the amount of variation (see Fig. 9) exclude the previous explanation. Figure 10 shows the map of the available water capacity (AWC) in the Salzach basin on a lower scale and the distance-factor function for standard deviation of AWC. On the map red boxes highlight spots within the basin comprising much higher AWC-values than its surrounding areas. These spots are also visible in the distance-

factor function. For the separated basin (red line) shows that the peaks are after the separation still visible, although the basic





heights of the line has been lowered. This observation can only be interpreted that the occurrence of such soil enclosures are a limiting factor for the reduction of heterogeneity with the ACS-algorithm. Though it does not restrict its applicability.

### 4.3 Dependence on basin structure

In the previous section we concluded that the shape of Mulde basin in combination with the present surface slope values caused
a decrease of performance. Additionally the arrangement of pore volume values in the Salzach catchment caused another decrease of performance. This conclusion brings up a fundamental question: is it the value-range of the considered characteristics or the spatial arrangement/basin shape that causes the issue? In other words: if we could examine the same basin with another set of values, would the outcome, i.e. the number of sub-basins, zonal extent and the performance criteria change? The problem is that no basin is like the other and even parts of the basin offer different structures and shapes than the entire
basin. Therefore it is comparably hard to draw conclusion to a single cause.

To overcome this issue we performed, what we refer as, a "resampling experiment". The intention is to examine the same basin shape and spatial arrangement with a different set of values (just like a time series analysis). Therefore a quantile-exchange of values has been performed.

Due to their similar size the basins of the Mulde and Salzach have been chosen for resampling. First we took the maps for
AWC of both basins (Salzach Fig.1, Mulde not shown, but spatial organisation is analogue to TPV, value range from 51 to 471 mm) and calculated an empirical distribution function for each basin. Subsequently the AWC values were replaced with their respective empirical quantile level. Finally the distribution functions were exchanged (Mulde to Salzach and vice versa) and the quantile levels were replaced with the exchanges distribution function quantiles. Results are shown in Fig. 11. We repeated this procedure with the DEMs as the source data for surface slope.
With this, admittedly bit confusing, resampling procedure we figuratively relocated the Mulde basin in a steep alpine environment with diverse soils and the Salzach basin is now equipped with mid-range mountainous heights and more homogeneous soil. Now we are able to assess the same basin shape and spatial arrangement with different (natural reasonable) range of parameter values.

The ACS-algorithm has been applied to the resampled values of pore volume and surface slope. Ascertained sub-basins, zones
and distance-factor functions of the respective standard deviation are shown in Fig. 12, performance values are tabulated in Tab. 2. In case of the Mulde basin the outcome does not change significantly. Ascertained sub-basins (number and shape) as well as zonal extent are very similar to the original results, performance values are stable. Note that the distance-factor function for $\sigma(AWC)$ has a very different shape than in the original basin (Fig. 4). Although the number of sub-basins ascertained for resampled slope in the Salzach basin has decreased from 38 to 5, the extent of zones (all three) as well as the overall
performance remains the same.

Application on AWC in Salzach basin shows a change in performance solely. While the total reduction decreases, the remaining variance above the threshold is 10% lower than in the original basin. This is also visible in the distance-factor function of $\sigma(AWC)$, note in contrast to Fig. 10 the missing peaks close to the outlet and in the more distant parts of the basin.



This result is explainable by the missing enclosures which have been transmitted has high values to the Mulde basin. Here, the allocation of the highest values is not organised in enclosures but in the upper-middle of the catchment (see Fig. 11) following a stream-orientated pattern. The peak of standard deviation is clearly visible in the distance-factor function of the unseparated resampled basin in Fig. 12, but the techniques of the ACS-algorithm are able to encompass this structure. Hence, the reduction

of variance remains on the same level.

In conclusion to this analysis we can state that the actual spatial arrangement, or more specifically its spatial organisation, defines the outcome of the algorithm. Since this was our intention, this a positive result. As a limitation we identified patterns (in this case soil patterns) that do not follow the co-evolutional structure of a basin (between soil and streams) (Blöschl et al., 2013) cannot be captured satisfactory by the proposed algorithm. Furthermore, a spatially homogenous variation structure of

catchment characteristic, independent from its actual amount of variation, is also complicated to assess with the ACS-algorithm. However, we could show that the proposed algorithm works well for catchment characteristics that offer wide range patterns (like soil properties) (In the supplement to this article we substantiate this statement by applying the algorithm to hydraulic conductivity data, results are in agreement to results of pore volume)

## 5 Method application

In the previous sections we have shown how the algorithm is working, how its outcomes are caused and what information about the basins we gain from its application. But its usefulness and its benefits are still unknown. We will address this topic in the following two sub-sections. In comparison to a common subdivision, we will first evaluate its reduction of variance and second show its benefits for the design of a hydrological rainfall-runoff model.

### 5.1 Comparison to gauging networks

The most common subdivision scheme is based on the available gauging network. On the one hand due to calibration requirements and on the other hand as only hind to a reasonable partition of the catchment. Obviously, existing gauging networks are a result of multiple considerations and requirements, e.g. of water management issues. In some parts of the basin it tends to be denser than required to catch the natural heterogeneity within a river basin, but other hydrological aspects (e.g. scale problems) are not considered in network design sufficiently. A comparison of sub-basins which are defined by

hydrological networks with the results of the ACS algorithm will show differences in the number of separation points. With the aim to reduce the variance such a comparison delivers hinds to decision makers to locate new gauges or (in our case) information about the usefulness of the ACS algorithm (Please keep in mind that the usefulness for the decision maker is limited by the informational value of the specific catchment characteristic for runoff generation processes.)

Two benchmark subdivisions were established:

1.   A subdivision based on the gauging network. It will be compared to the obtained ACS-basins (without zones).





2.  A subdivision based on the gauging network with an additional zonal partition by land cover, based on the suggestions by Lindström et al. (1997)(Additional third zone "Rock/Bare soil" has been introduced to account for alpine structures in the Salzach basin). Will be compared to full outcome of ACS.

Gauging networks and defined land cover zones are shown on the left of Fig. 13. The distance-factor function of standard deviations for pore volume (mid of Fig. 13) and slope (right) are shown as well as results of ACS-subdivisions. Performance measures are shown in Tab. 3.

Let us first focus on pore volume. In the distance-factor function it is visible that the red-line, representing ACS-results, is at nearly all distances below the blue (rep. gauges) and the green (rep. gauges & land cover) line. The impression of ACS-advantage is confirmed by the performances measures in Tab. 3. The total reduction $\alpha_1$ (Eq. 8) exceeds values for gauges & land cover (bench-2) by a factor between 1.3 and 1.8. Moreover the remaining variance above the threshold $\alpha_2$ (Eq. 9) is 16% to 45% lower. Please note that this advantage is realised with a significantly lower number of subdivisions. Leading to the conclusion that the proposed algorithm is a more effective tool to lower the heterogeneity of pore volume data, or any catchment characteristic that has a similar spatial organisation (see Supplement).

Just as in Section 4, results for slope offer a different impression and quality. In the distance-factor functions (Fig. 13) we can see that all lines are on an equal level and a clear advantage for any of the partition strategies is not visible. But, if we look at the performance results in Tab. 3, we can see that without zonal classification the gauging network has an advantage to ACS-results. Especially the sub-divisions in the Mulde basin are ineffective. However, with the introduction of zonal classification the performance values become more alike. Ranges for $\alpha_1$ cover (with exception the Regen catchment) the same range of 8% to 17 %. But the ranges of remaining variance are more diverged (22-77% ACS and 30-54% bench-2) and point out that the benchmark partition captures the heterogeneity of slope at least at some points of the basin better than the ACS.

This result is, again, caused by the fragmented nature of surface slope values. As we could show the gauging catchments solely does not capture the heterogeneity of surface slope, so we can draw its performance to the zonal classification which can be described as small scale distributed and fragmented.

## 5.2 Modelling application

Our model of choice is the $HBV_{96}$-model (Lindström et al., 1997), due to the fact that we already used its recommended catchment zonal classification scheme, its obvious ability to incorporate zones and, after all, due to its common usage.

The $HBV_{96}$-model is a semi-distributed conceptual model. Each sub-basin has one lower and upper ground water storage responsible for slow and fast runoff generation. On top of these storages an arbitrary number of zones can be placed. Each zone has an individual atmospheric-, interception-, snow- and soil water-routine. In the original concept four types of zones are possible: Field, Forest, Lake and Glaciers. The latter has not been used in this study so we pass its description. Field and Forest zones are conceptually identical; their purpose is to provide different parameters for different land cover. Lake zones do not comprise any soil routines. All precipitation on this zones is handed directly into the lower ground water storage. Since the modelling application is meant to be short, please see (Lindström et al., 1997) for further information about the model.



Our application is the Mulde catchment due to good data availability. Daily mean discharge, precipitation sums, temperature means and sums of potential evapotranspiration time series from 1951 to 2011 are available for 39 gauged sub-basins (discharge data for 20-39, dependent on time window).

Two spatial model setups were employed for application, both are shown in Fig. 14. The left of Fig. 14 shows the bench-2

partition as we already used it in the previous section, based on gauging network, heights and land cover. On the right of Fig. 14 our proposed subdivision of the catchment, based on gauging network and pore volume is shown. The gauging network has been used as a pre-partition of the basin and the ACS refined the sub-basin density and defined zones for all sub-basins. We chose pore volume as catchment characteristic whose spatial organisation to incorporate into the model structure. Our decision is based on the fact that ACS worked better for pore volume than for slope and that information about storage capacities seemed

to be a more valuable information for a conceptual (storage based) model.

Beside the incorporation into the model structure we are able to use information about the catchment characteristics in the calibration process. Each (ACS) sub-basin possess three zones, each has an individual average value for pore volume. Since we minimised the heterogeneity of the respective characteristic we are able to assume a uniformly distribution of this value for the entire zone. Now, an automated calibration routine gains profits from these information and we are able to reasonably

couple parameters of the zones by their respective average of the characteristic.

Say, each zone comprises a parameter called $X$. An algorithm, like the BOBYQA-algorithm employed in this study (Powell, 2009), offers in each iteration step a guess for this parameter $\dot{x}$. The parameter guess is then transformed to the zonal-parameter by the zone characteristic $E_{Zone}(C)$ normalized by the average value of the entire basin $E(C)$ and a linear coupling parameter $k_X$:

$$X_{Zone} = \frac{E_{Zone}(C)}{E(C)} \cdot k_X \cdot \dot{x} \qquad (10)$$

Please note that the coupling parameter is subject to the calibration process itself and bound to a single parameter. In this study we coupled 6 zonal-parameters that are related to soil properties. This coupling-scheme (comparable to studies by Gharari et al., 2014) has been applied to the benchmark additionally, here $E(C)$ in Eq.10 is omitted and applied only to field zones, while forest zones use $\dot{x}$ unchanged. Since the coupling of the benchmark subdivision is not based on any process assumptions we

additionally applied a free parameterisation, where all parameters within the each sub-basin and zone are can be used for model fitting.

As already mentioned we performed calibration by means of the BOBYQA-algorithm (Powell, 2009), progressively from the headwaters towards the outlet of the basin. Both benchmark calibrations employ the same spatial setup with 38 sub-basins and an average of 30 zones per sub-basins. Due to the different parametrisation strategies a different but equally high number of

parameters are subject to the calibration (see Tab. 4). The ACS-structure employs 44 sub-basins but only 3 zones (except 2 sub-basins only 1), giving 51 parameters per sub-basins. Compared to the benchmark calibrations this number 6 to 9 times lower.





After the calibration (time period 1995-2006) we evaluated model performance in three validation periods. Two in direct (temporal) neighbourhood to the calibration period and the last at the very beginning of the time series. Model performance has been calculated as the average Nash-Sutcliffe-Efficiency (NSE) (Nash and Sutcliffe, 1970) of all gauges and is tabulated in Tab. 5. Results show that ACS-parametrisations are superior in all cases. Its increase in performance ranges from 17-52%

in comparison to the free- and 11-21% to the coupled benchmark parametrisation.

Beside this "lumped" evaluation we compared the performance of the models at each gauge in each period. Comparison of NSE for coupled models are shown in Fig. 15, for ACS and free-benchmark model in Fig. 16. We can see that the individual performances offer the same conclusion as the lumped performance, though some results are better for benchmark models (both parametrisations). To be more precise, in case of coupled models 20 points (rep. a single gauge in one of the time periods)

are below equivalency (rep. a better performance of the benchmark model) and in case of the free-benchmark model 12 points. Representing 15% or 9% of evaluated cases.

To conclude we have to ask: What is the result of this modelling study? Obviously, we could improve modelling performance. In agreement with findings in literature we could prove that additional information relevant to hydrological processes improve model performance (Finger et al., 2015; Li et al., 2015) and, moreover, we can posit that the spatial organisation of catchment

characteristics (in this case pore volume) is a relevant information. The latter conclusion is drawn from the fact that the ACS-model offers a similar (or superior) model performance to the coupled model though it comprises less model parameters.

# 6 Conclusions

The intention of this study was to assess the spatial organisation of catchments and their characteristics as well as to evaluate the benefit we can gain from this information for the use in conceptual, semi-distributed hydrological models. At first we

proposed the distance-factor function to assess the interaction of an arbitrary catchment characteristic with the flow path. Graphical representations of this function are able to make the heterogeneity of the considered characteristic visible. We further build an algorithm on this proposed function, mainly focusing on factor-functions of standard deviations, with the objective to lower the heterogeneity of the respective catchment characteristic. The proposed ACS-algorithm utilises three different techniques two lower the heterogeneity that are deduced by the main sources of heterogeneity visible in natural catchments.

The outcome of the algorithm offers a spatial subdivision of the catchment, comprising a minimum of standard deviation of the respective characteristic.

After the introduction of these methods we performed an extensive test of the ACS-algorithm. First we tested model functionality and limits of application on four different basins. We evaluated obtained spatial patterns of the outcome with visible spatial patterns in the basins. Furthermore we compared the reduction of variance for different characteristics and

basins. Next we evaluated the usefulness of the obtained results. On the one hand we compared the variance reduction to a benchmark separation and on the other hand we merged our results to a semi-distributed hydrological model. The modelling study showed the benefits we can draw from the incorporation of spatial organisation of the basin.





We could show that the distance-factor function is a useful tool to detect non-random spatial patterns and the interaction of catchment characteristics with the flow path. Moreover, it is capable to detect anomalies in the structure of the catchment, like spots of different soil types that do not follow the co-evolutional structure of the basin.

The proposed ACS-algorithm showed good results for different catchment forms, sizes and patterns. Heterogeneity of characteristics with spacious patterns (like soils) are beneficial for the application of ACS algorithm, in terms of variance reduction. For more fragmented characteristics (like surface slope) that offer a small-scale but spatially equally distributed heterogeneity the algorithm certainly offers a subdivision and zonal classification, but the variance reduction is comparable to common approaches for basin subdivision. Moreover we detected the general basin shape as influential for the outcome of the algorithm. Although this is quite obvious, we identified that basins arranged in along a single axis (like a strict south to north orientation) where the variance of catchment characteristics are highly correlated to the distance to the outlet, are harder to assess for the proposed algorithm.

Our future work will focus on two topics: On the one hand we have to further improve the subdivision algorithm. At this point we are able to assess the structure of a single characteristic, while it is highly desirable to consider multiple characteristics. Moreover we have to develop methods to encompass soil enclosures and fragmented characteristics. Although the latter problem might lead to the well-known HRU-concept. We have to elaborate if such development is desirable. On the other hand we will address the value for catchment similarity studies. Following intentions by Mesa and Mifflin (1986) who suggested the width-function as an indicator for catchment similarity, it is questionable how results of the distance-factor function can be used to characterise similarity.

## 7 Code & Data availability

Python code and Toolboxes for common GIS-Software products of the proposed ACS-algorithm are available at https://github.com/HenningOp/ACS. Spatial data used in this study (DEM, Soil data, drainage points) are made available as well.

## Appendix A

In this Appendix details to the proposed ACS-tools are given for further understanding of the algorithm. Each tool will be addressed separately.

## Detachment tool for low variance regions

Low variance regions have no need for further subdivisions, hence they are detached from the rest of the basin. Since the exact allocation of these regions is known, all cells within can be defined as target area T (hatched in Fig. 6a). Remaining cells are drawn together as non-target area NT. If one random point of the basin is selected as possible separation point (SP) and its





watershed is calculated the set of points belonging to the watershed, or sub-basin, of SP, BSP is obtained. The calculated watershed BSP covers parts of T and NT and hence a coverage rate can be calculated as the proportion of the cardinalities of the intersections and their respective superset:

$$O_{SP} = \frac{\left| B_{SP} \cap T \right|}{\left| T \right|} - \frac{\left| B_{SP} \cap NT \right|}{\left| NT \right|} \to \max \tag{A1}$$

The objective of a detachment O is to find a separation point (SP) whose basin BSP covers a maximum of T and a minimum of NT. Please note that for regions located at the outlet of the basin or at its upstream boundary only one SP will be defined. Possible SPs are assumed to be allocated at the transit of the main stream from T to NT, or vice versa. An iterative search returns the coverage values O and the highest value is selected as SP, defining a new sub-basin. In the upper part of Fig. 6a concluded separation, as well as the rejected SPs of the iteration (hollow points) are shown.

**Pruning at confluences/branches**

To identify branches the distance-factor function of flow accumulation (i.e. cumulative number of cells draining into a cell) (FAcc) is examined. FAcc indicates the contributing drainage area to each stream cell, hence discontinuities in the distance-factor function indicate confluences of streams (See Fig. A2 for example of distance-factor function).

Beginning at the Outlet (zero on the x-axis) two features are visible: a slowly decreasing line of high FAcc values, representing
the main stream at the outlet, and a noise-like smaller range of FAcc values close to the abscise, caused by the smaller tributary streams and contributing areas. To identify major tributaries this noise has to be removed. We assume that in the first distance class the disparity between main rivers and contributing hillslopes is most distinct. Within the first class a k-Means cluster analysis is carried out to divide high and low FAcc values. Threshold value $\tau_S$ is determined as:

$$\tau_S = \min_c \left[ \max_{c1} \left[ FAcc \right]; \max_{c2} \left[ FAcc \right] \right] \cdot \left( 1 - \frac{\gamma}{10} \right) \tag{A3}$$

Where $c$ indicates the clusters and $\gamma$ is the reduction order and by default 0. The algorithm will start with the default value for $\gamma$ and searches for the first branch in upstream direction. If no branch is found, order $\gamma$ is increased by 1. The maximum order is set to 10. Please note that the higher $\gamma$ is set, the lower the threshold gets and more FAcc-values remain for analysis. The routine identifies the coordinates of the branch inducing the drop in FAcc values (see Fig. A2, for example at distance ≈ 80 km).

Drainage points of major streams, or major stream branches are identified likewise. Before the objective function is called, prevailing FAcc values in the basin are checked. If the FAcc value of the tributary stream is higher than the threshold value $\tau_R$, a subdivision is performed. This parameter is calculated as percentage of the maximum FAcc value in the entire watershed (e.g. 5%) once at the initialisation of the algorithm.





**Zonal classification**

Iterative search for the optimal zonal classification involves three parameter: reduction of straher order $s_R$, distance from stream $o$ and heights quantile $h$. The maximum Strahler order within the considered sub-basin $mS$ also involved but a constant value. Initial values are: $s_R = 0$, o = 0 and h = $0/h_{ite}$ (note that $h_{ite}$ is a required, user defined parameter > 1.). In each iteration step one

of the three parameters is increased until its maximum value ($s_{R;Max} = mS$; $o_{Max} = 5$; $h_{Max} = h_{ite}$) creating a different composition of zonal extent.

$s_R$: Controls which cells within the basin are potential "*close to stream*" (CTS) zones. All stream cells of Strahler order greater/equal to $mS$-$s_R$ and all non-stream cells draining into a stream cell fulfilling this requirement are potential CTS-zones.

$o$: Defines the width of CTS-zones. All potential CTS-cells with $x_H ≤ o·\Delta o$ (as defined in Sec. 3) are confirmed as CTS-cells,

all remaining cells of the sub-basin as "*transition*" (TS)-cells.

$h$: Controls the threshold used to define "*high elevation*" (HE) zones. An empirical distribution function of heights (taken from the input DEM) of all FFS-cells is calculated. The height threshold $\tau_H$ is then calculated as the $h/h_{ite}·100$ [%] quantile of the empirical distribution function. All cells with an assigned heights > $\tau_H$ are filed HE-zones, all remaining cells are confirmed as TS-cells.

After each iteration the averaged, distance-based standard deviation (Eq. 6) is calculated. Parameter combination giving the lowest $\sigma_S(C)$ is chosen as result.

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





**Table 1: Results of applications of *ACS*. Number of ascertained sub-basins, normalized reduction of standard deviation**

| Catchment | Pore volume | | | Slope | | |
|---|---|---|---|---|---|---|
| | No. of Basins[-] | $\alpha_1$ (Eq. 7) [%] | $\alpha_2$ (Eq. 9) [%] | No. of Basins [-] | $\alpha_1$ (Eq. 7) [%] | $\alpha_2$ (Eq. 9) [%] |
| Mulde | 38 | 54.3 | 10.4 | 30 | 8.2 | 77.9 |
| Main | 59 | 65.2 | 0.9 | 22 | 17.7 | 54.3 |
| Regen | 17 | 62.5 | 13.5 | 24 | 28.0 | 22.1 |
| Salzach | 24 | 48.5 | 25.6 | 38 | 15.1 | 56.0 |

**Table 2: Normalized reduction of standard deviation for resampled basins**

| Catchment | Pore Volume | | Slope | |
|---|---|---|---|---|
| | $\alpha_1$ [%] | $\alpha_2$ [%] | $\alpha_1$ [%] | $\alpha_2$ [%] |
| Mulde (res) | 57.9 | 12.8 | 8.5 | 82.6 |
| Salzach (res) | 38.7 | 10.5 | 14.7 | 58.5 |

**Table 3: Normalized reduction of standard deviation for sub-basins based on gauging network, ACS-basins and gauges and land cover**

| Catchment | Pore volume | | Slope | | No. of Gauges |
|---|---|---|---|---|---|
| | $\alpha_1$ [%] | $\alpha_2$ [%] | $\alpha_1$ [%] | $\alpha_2$ [%] | |
| *Gauging network* | | | | | |
| Mulde | 24.2 | 53.4 | 9.4 | 69.9 | 40 |
| Main | 41.9 | 26.2 | 14.0 | 44.0 | 46 |
| Regen | 21.1 | 74.6 | 10.1 | 57.9 | 20 |
| Salzach | 30.3 | 48.8 | 9.6 | 68.5 | 33 |
| *ACS-basins only* | | | | | |
| Mulde | 32.8 | 45.3 | 0.0 | 100.0 | 38/30 |
| Main | 50.7 | 8.5 | 9.2 | 72.9 | 59/22 |
| Regen | 40.9 | 35.9 | 14.1 | 52.8 | 17/24 |
| Salzach | 40.6 | 24.6 | 9.3 | 73.6 | 24/38 |
| *Gauging network & land cover* | | | | | Occ. zones |
| Mulde | 35.3 | 35.0 | 14.5 | 54.8 | 2 |
| Main | 48.9 | 17.7 | 19.8 | 26.2 | 2 |
| Regen | 33.2 | 59.4 | 19.1 | 27.4 | 2 |
| Salzach | 38.4 | 50.1 | 21.6 | 30.8 | 3 |



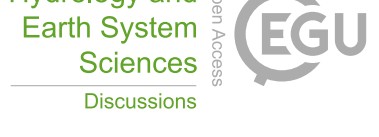

**Table 4: Parameter quantities**

|                      | Benchmark *Free* | Benchmark *Coupled* | ACS *Coupled* |
|----------------------|-----------------:|--------------------:|--------------:|
| Sub-basins           | 38               | 38                  | 44            |
| Zones per Sub.       | ~30              | ~30                 | 3             |
| Parameter (total)    | 19562            | 12198               | 2244          |
| Parameter (per Sub.) | ~495             | ~321                | 51            |

**Table 5: Nash-Sutcliffe Efficiencies of Benchmark and ACS-model**

| Simulation (Start-End) | $NSE_{B;Free}$ [-] | $NSE_{B;Coupled}$ [-] | $NSE_{ACS}$ [-] |
|------------------------|-------------------:|----------------------:|----------------:|
| 1995 - 2006 (C)        | 0.678              | 0.659                 | 0.792           |
| 2007 - 2011 (V)        | 0.524              | 0.570                 | 0.622           |
| 1984 - 1995 (V)        | 0.496              | 0.516                 | 0.607           |
| 1951 - 1961 (V)        | 0.433              | 0.568                 | 0.660           |





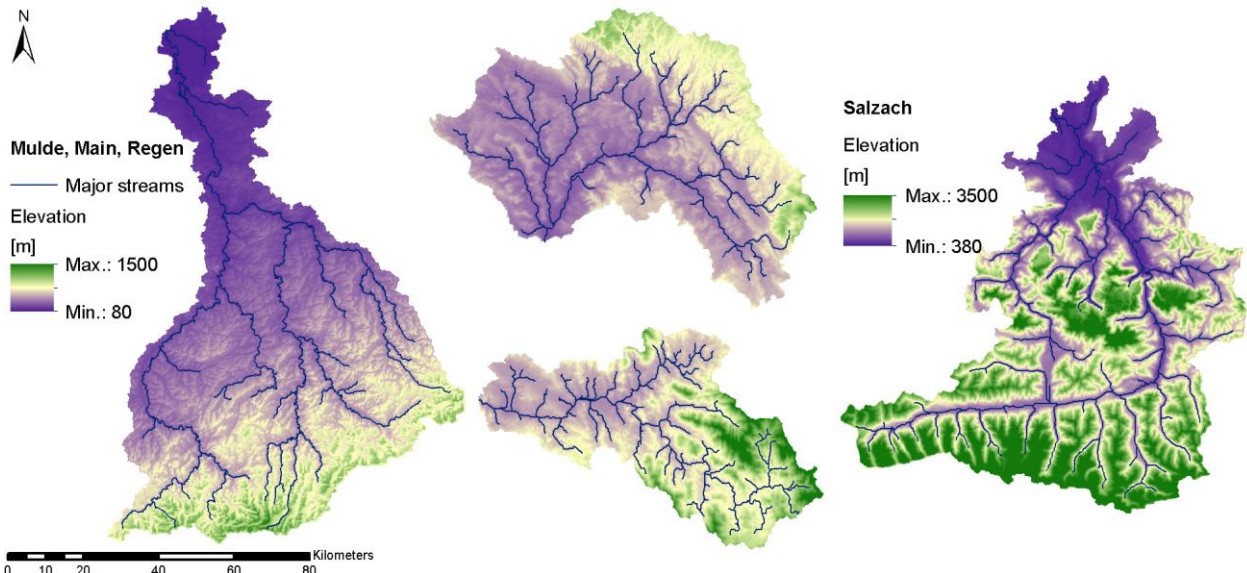

**Figure 1: Digital Elevation models of the Mulde (left), upper Main (mid upper case), Regen (mid, lower case) and the Salzach (right)**

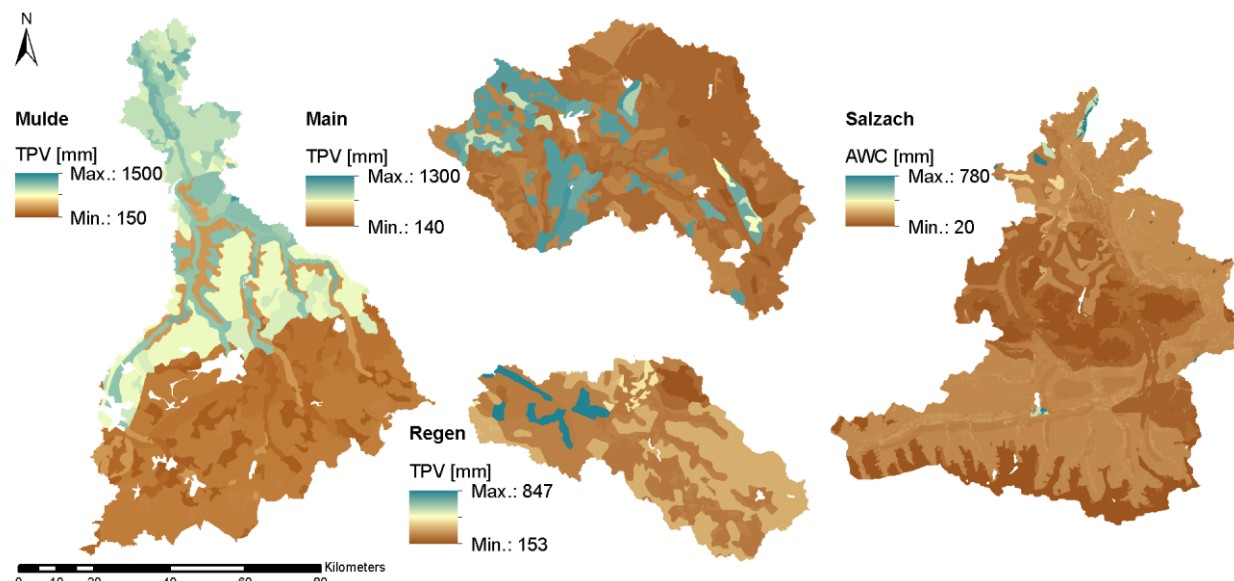

5 **Figure 2: Values of total pore volume of the Mulde (left), upper Main (mid upper case), Regen (mid, lower case) and AWC of the Salzach (right)**




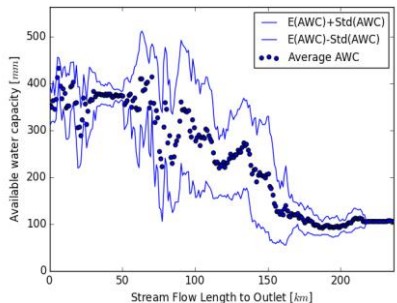

**Figure 3: Distance-factor function of AWC in the Mulde catchment**

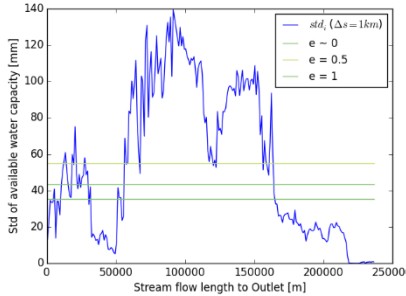

**Figure 4: Distance-factor function of $\sigma$(AWC) and threshold values $\Omega$ for different values of $e$, in the Mulde catchment**

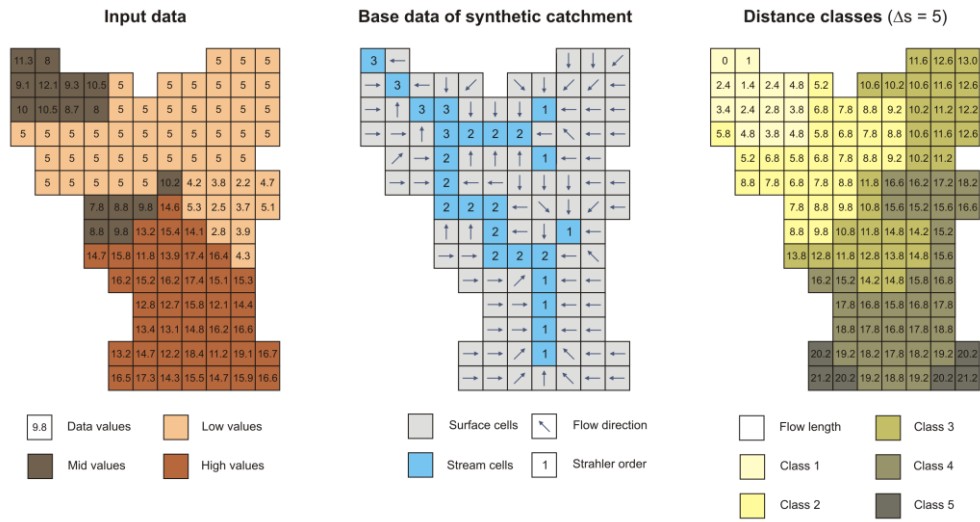

**Figure 5: Example input data (right), flow direction and Strahler order (middle) and distance data and –classes (right)**





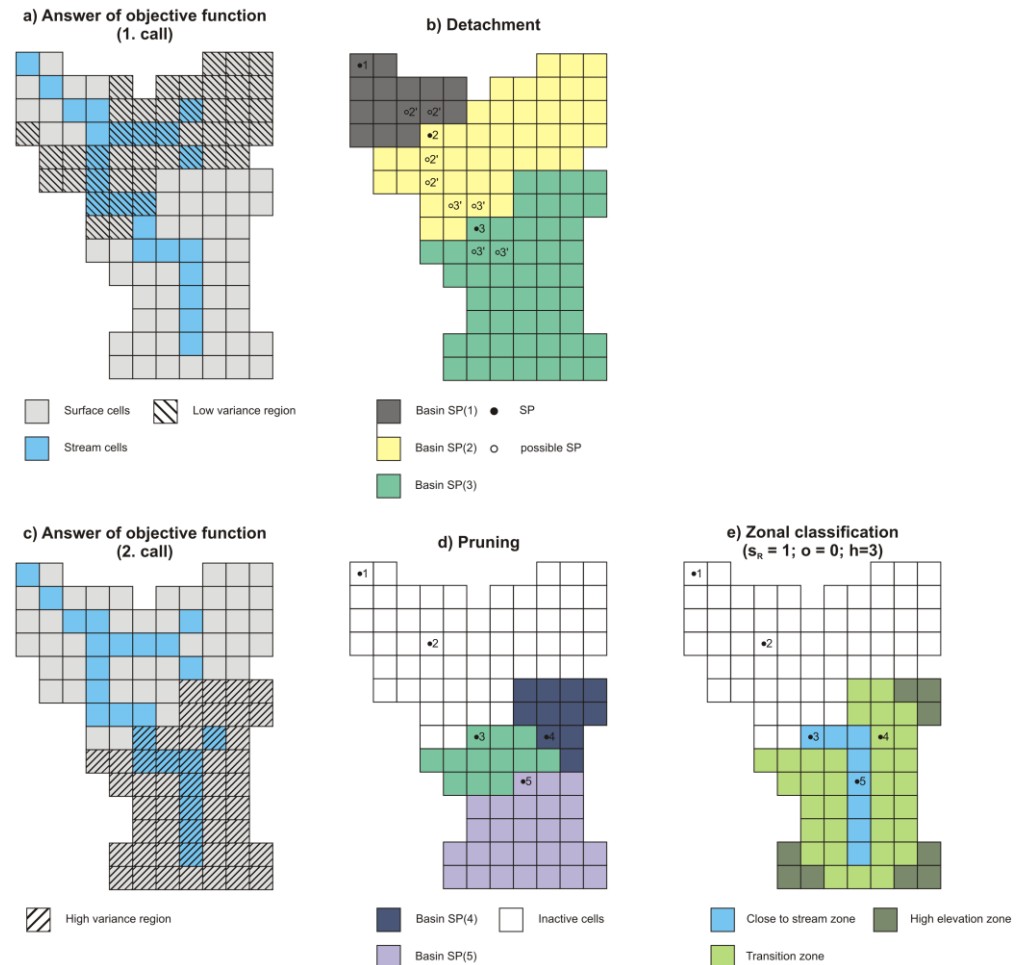

**Figure 6: a&c) Answers of the objective function; result of b) Detachment, d) Pruning and e) zonal classification in the synthetic catchment**




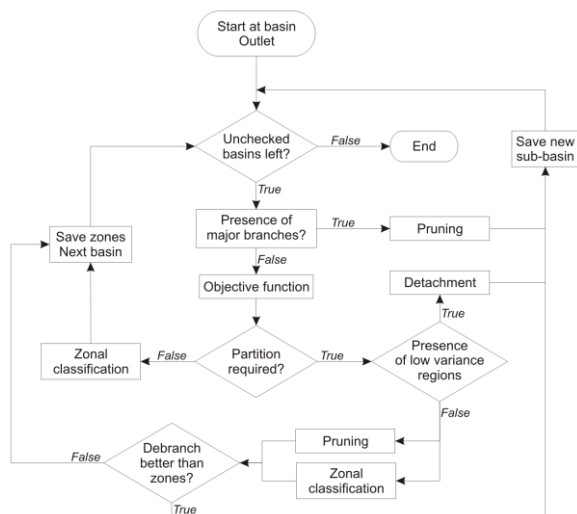

**Figure 7: Sequence of the ACS-algorithm**

**Figure 8: Results of ACS application for catchments of the Mulde and Regen, sub-basins based on pore volume (left) and slope (right). Comparison of $\sigma_U(C)$ and $\sigma_S(C)$ for each application (red and blue lines).**



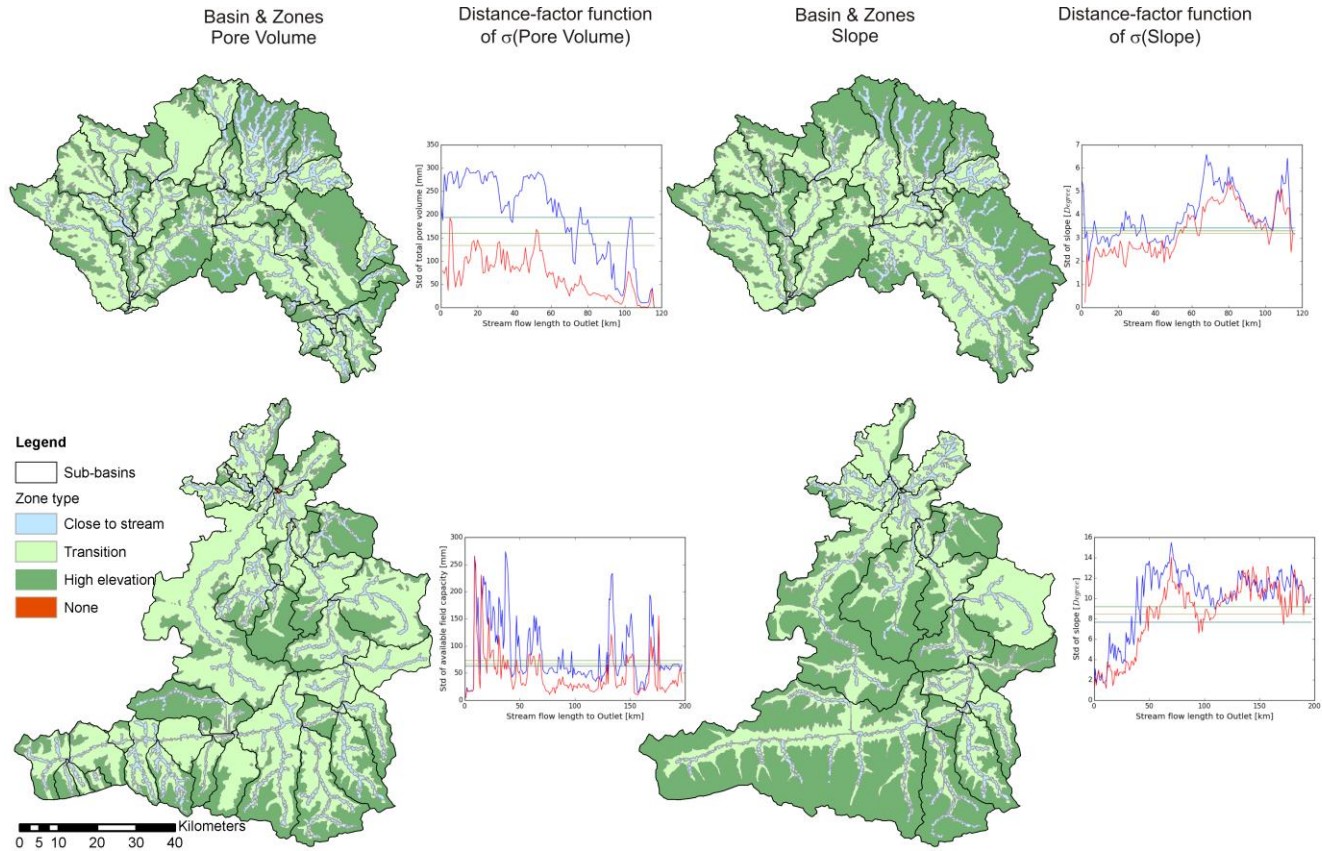

**Figure 9: Results of ACS application for catchments of the Main and Salzach (from top to bottom), sub-basins based on pore volume (left) and slope (right). Comparison of $\sigma_U(C)$ and $\sigma_S(C)$ for each application (red and blue lines).**

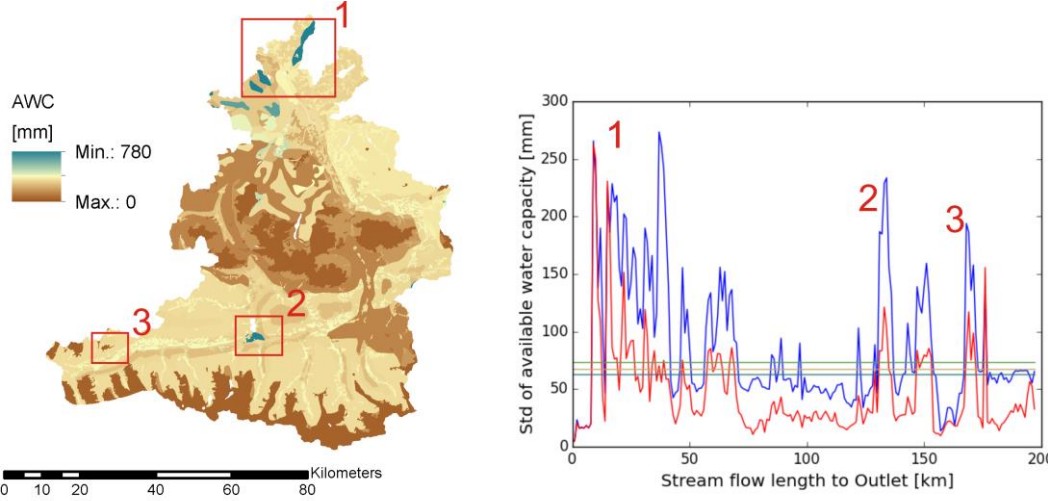

5    **Figure 10: AWC of the Salzach catchment and the distance-factor function of $\sigma_U(C)$ and $\sigma_S(C)$. Red marked and numbered areas incorporating high value enclosures**





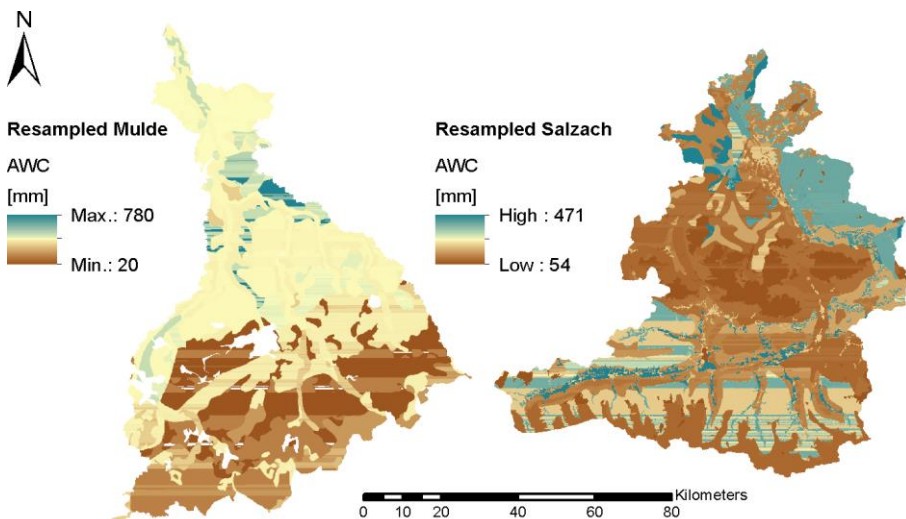

Figure 11: Resampled AWC values for Mulde and Salzach catchment





**Figure 12:** Results of ACS application for resampled catchments of the Mulde and Salzach (from top to bottom), sub-basins based on resampled pore volume (left) and slope (right). Comparison of $\sigma_U(C)$ and $\sigma_S(C)$ for each application (red and blue lines).





**Figure 13: Subdivisions based on gauging network & zonal classification and distance-factor functions of σ(pore volume) and σ(slope) (left to right) for catchments of the Mulde, Main, Regen and Salzach (top to bottom)**





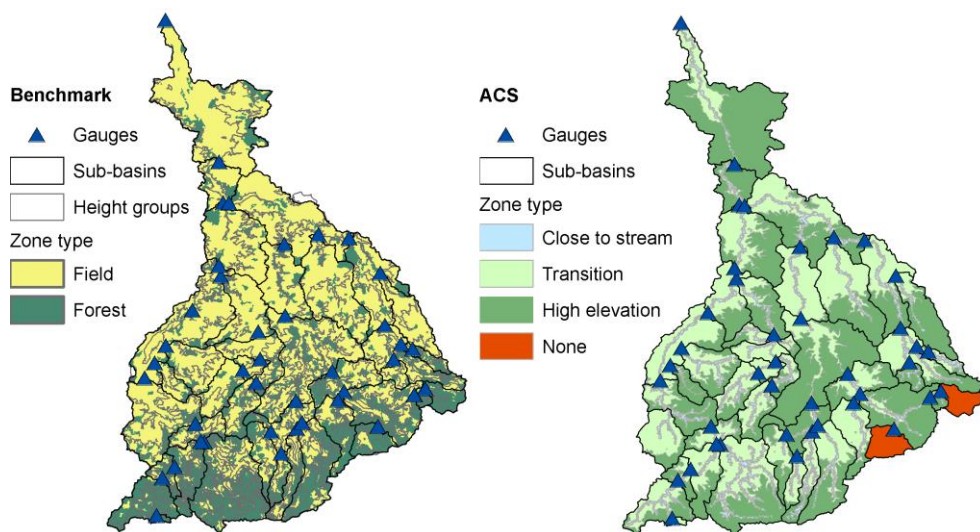

**Figure 14: Spatial structures for HBV₉₆-Model: (left) ACS-basins & zones; (right) gauging network, landuse & heights**

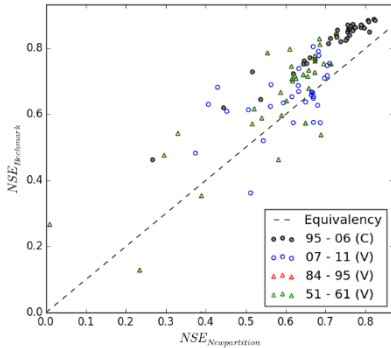

**Figure 15: Nash-Sutcliffe Efficiency of ACS-based model and benchmark model, coupled parametrisation**

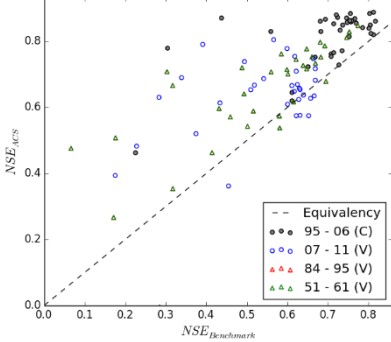

**Figure 16: Nash-Sutcliffe Efficiency of ACS- based model and benchmark model, free parametrisation**



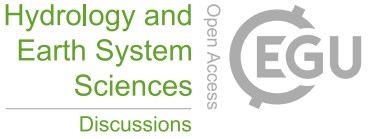

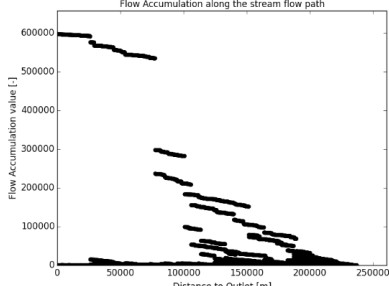

**Figure A2: Distance-factor function of Flow Accumulation in the catchment of the Mulde**