# Peer review of "A method to employ the spatial organisation of catchments into semidistributed rainfall-runoff models"

_Hydrology and Earth System Sciences, 2017_

## Referee Comment (RC2) · Anonymous Referee #2 · 26 May 2017

**Review of: "A method to employ the spatial organisation of catchments into semi-distributed rainfall-runoff models" by Henning Oppel and Andreas Schumann**

The paper presents an interesting effort to utilise spatial organisation of catchment properties to improve the division of a catchment into sub-catchments. The analysis appears promising in the exemplified application on four German catchments using distributed information of pore volume and slope. The obtained variance reduction proves the applicability of the developed method and the associated tools. However, the paper is not recommendable for publication in its present form for more formal reasons.

First problem is that some basic mathematics is unclear, e.g. eqs. (1) and (2). The summation variable in (1) needs clarification. $x$ is running from $i \cdot \Delta s$ to $(i+1) \cdot \Delta s$, but there is no indication of the range of $x$-values to be applied. In (1) the mean value $E(C)_i$ is established, but in the calculation of the standard deviation in (2) the applied mean value is denoted $E(C)_x$. Such basic formulas must be better presented and explained.

Second problem is the awkward English, which is below an acceptable standard for a high quality journal. There are numerous grammatical errors (punctuation; wrong tense, wrong words, missing words, bad construction of sentences etc.), which is very disturbing for grasping the real content of the paper.

The authors are encouraged to critically revise the paper and resubmit.

---

## Referee Comment (RC3) · Anonymous Referee #3 · 8 Jun 2017

This study presents a new method for analyzing the spatial organization of the catchment properties relevant for rainfall-runoff modelling. The method is applied for subdividing the catchment in sub-basins with minimum internal variance of the analyzed properties. This can be relevant for reducing the uncertainty in parameter identification of semi-distributed models, which discretize catchments in units to be modelled as lumped. It can be also exploited for designing stream gauge networks. The method is based on a "distance-factor function", which describes the spatial variability of a given catchment property with respect to classes of flow path length. The performance of the proposed method is evaluated on four large catchments in Germany and Czech, by evaluating the "pore volume" and the terrain slope. A significant reduction of the

variance is achieved for the pore volume. HBV model is also applied for evaluating the impact of the proposed catchment discretization on the simulated catchment response. The results show a higher Nash-Sutcliffe Efficiency of the discharge simulated with the proposed catchment discretization with respect to two other benchmark methods.

The proposed methodology appears to be promising, but the manuscript needs to be improved a lot.

The algorithm is not clear. Many equations are not formally correct. Equation (3) describes a threshold value for the variance of the catchment property. I do not understand the theoretical motivation behind the definition of this threshold value. The exponent parameter "e", described as non-linearity factor, is not mentioned in the following paragraphs. Is this a calibration parameter? What was the value applied in the examined case studies? It is not clear why it appears only in the numerator and not in the denominator. Equation (3) is not a weighted sum of the class variances if "e" is not equal to 1. Anyway, the motivation to weight the variances by their differences with respect to the maximum variance is quite unclear.

The readability of the paper is very poor. I needed to read it a few times to get a clear picture of the paper content. A revision by a native speaker can improve the paper. However, I also think that the structure of the paper should be revised. The sections dedicated to the description of the algorithm should be improved by adopting a more rigorous mathematical formulation. Many paragraphs are verbose, with comments that are not relevant for understanding the proposed methodology. The appendix does not help either.

The evaluation based on the comparison of the HBV model output performances is not convincing. The outcomes depend on the adopted calibration procedure, which is not clearly described. An analysis of the parameter uncertainty would be more valuable.

The evaluation of the proposed methodology could be limited to the analysis of the variance reduction, while the analysis of its impact on the simulated catchment response

could be discussed in a second manuscript.

Reducing the length of the paper, by focusing on the essential aspects, may improve the impact of the research.
* * *

---

## Author Comment (AC1)

**Response to Reviewer Comment #1**

We would like to thank you for your comments and very constructive suggestions. Your feedback on the manuscript is very valuable, and it will help us to improve the manuscript.

Response to general comment on model comparison is given below under Comment no. 13. Criticism of linguistic quality caused us to repeat and extent our proof-reading procedure. The linguistic revision is not yet completed and will be addressed separately in an additional response. Here we will focus on the content-related comments by the referee.

Response to detailed comments:

1. "Page 3, line *20* 30: The computational demands of semi-distributed models with polygon based delineation is nowadays often negligible."

   We agree that the computational demand to perform a single model run is often negligible, but this omits the demands for model calibration or extensive sensitivity analysis studies. However, the actual formulation does not address this subjects and has been reformulated as follows:

   "The aim of the proposed algorithm is set to offer a basin partition with a minimum of heterogeneity by a minimum of sub-divisions, i.e. to  *a unnecessary high number of sub-divisions and, hence, number of parameters* in cases of hydrological modelling."

2. "Page 4 and Fig. 2: Pore volume and available water capacity should be defined. For instance is with "total pore volume" the "porosity" meant? Usually these variables are given in volume percent? Here they are given in mm, why? Does this not require an information about the soil depth?"

   Our data basis provides the opportunity to assess soil properties in different depths. In previous projects, where this data has been applied, we experienced that the majority of soil storage is located in the upper zones of the soil layer. Namely a depth of up to 2 meters covers nearly all available soil storage. Why we applied the *[mm]* data and not the *[%]* is in this case due availability (data from previous project) than for any particular consideration.

   The employed TPV data is calculated for a soil depth of up to 2 meters, this information has been added to the manuscript, Legend in Figure 2 and Page 4 Line 18-19 have been revised:

   "To characterise the soil characteristics of the German catchments, a gridded soil data map from the German Federal Institute for Geosciences and Natural Resources (BÜK200) and CORINE land coverage data (CLC) (Bossard et al., 2000) were used. Pedo-transfer functions (Sponagel, 2005) were applied to transfer these information into gridded data about (available) water capacities (AWC) and other soil storage parameters as well as hydraulic conductivities, *both for a soil column of up to 2 meters depth*."

3. "Page 5, Lines 24-25: I was trying to picture the basin split into stripes. I found Figure 5 as an visual explanation. However, comparing Fig. 5 with Fig. 3 there should be much less points of the distance-factor function in Fig. 3 (only 5 points). Later I realised that Fig. 3 does not belong to Fig. 5. For demonstration purpose I would suggest providing a pair of Figures with distance classes and corresponding distance factor function using a few classes only. May be Fig. 5 needs to come before Fig. 3."

The enhance comprehensibility we follow your recommendation. The distance-factor function of the synthetic catchment has been created (see Figure below) and added to the manuscript as Fig. 4. The synthetic catchment is now introduced on Page 5, Line 30 as follows:

"Please note that w(x) is the non-normalised value of the area-function (Snell and Sivapalan, 1994). *To visualise the proposed function a simple synthetic basin with its stream network, distance-classes and an arbitrary characteristic are shown in Fig. 3. To keep things simple Fig. 3c shows the unified flow length (comprising $x_S$ and $x_H$) derived from flow direction data in Fig. 3b. As it can be seen in Fig. 3c the basin has been split into 5 distance classes. When Eq. 1 and 2 are applied to the shown data (Fig. 3a) the average and standard deviation are calculated within these five distance classes. The obtained distance-factor function is shown in Fig. 4.*
Figure  *4* shows the application of  *to real data in a meso-scale catchment, namely for AWC in the Mulde catchment.*"

[Figure]

**Figure 3: Distance-factor function of sample Data in synthetic catchment**

4. "Page 5, line 31: (Eq. 1 and 2)"

Already addressed in previous point.

5. "Fig. 4: Use the same units on the x-axis as in Fig. 3."

Figure 4 (now 5) has been revised.

6. "Fig. 5: Example input data are on the "left"!"

    Subscript of Fig. 5 (now 3) has been changed as noted.

7. "Page 12, Eq. 8: I would suggest to explain this equation in words. I also would suggest to exchange the sides of omega and sigma (even if mathematically not necessary) in both numerator and denominator."

    We suppose the suggestion to switch omega and sigma is indented to clarify the summation. Instead of switching the summands we added parenthesis for clarification. An explanation of Eq. 8 has been added to the manuscript on page 12, line 4:

    "Second measure $\alpha_2$ is intended to show cases where the total heterogeneity has been lowered significantly, but still remains above the objective. *Threshold $\Omega$ is used to formulate this objective. Finally $\alpha2$ is calculated as the difference between the threshold $\Omega$ standard deviation in the separated catchment $\sigma S(C)$, normalised by the difference between $\Omega$ standard deviation in the unseparated catchment $\sigma U(C)$:*

$$\alpha_2 = \frac{\sum_{i \in M(S)} \left( \Omega - \sigma_{S;i}(C) \right)}{\sum_{j \in M(U)} \left( \Omega - \sigma_{U;j}(C) \right)} \tag{8}"$$

8. "Page 12, line 15: What are cases with "negative outcome"? Do you mean insufficient variance reduction?"

    That is correct. Page 12, line 15 has been revised as suggested:

    "If we focus on the cases with  *insufficient variance reduction,* we are able to identify some limitations of the algorithm."

9. Fig. 8, 9 and 12: The distance factor functions are hard to read. Use vector graphics and/or large fonts and/or larger figures.

    Figures have been revised. Figures are now embedded in EPS format, fonts have been enlarged.

10. "Fig. 10: In the legend of AWC "max" and "min" need to be exchanged."

    Legend of Fig. 10 has been revised.

11. Fig. 11: When resampling of AWC for the Mulde river basin is shown also the original AWC map of the Mulde basin should be shown for comparisons. Why are there some kind of horizontal stripes in Fig. 11?

    The map of original AWC values in the Mulde basin is not shown in the manuscript, because the spatial arrangement of AWC and the shown TPV is identical (as mentioned on page 13, line 15) due to origin of these data grids. Both values follow the arrangement of the soil map, their only difference is the range of

values (also given in line 15), caused by different calculation formula (Sponagel, 2005). Since the manuscript already has a great number of figures and the information content of the AWC figure low we omitted it in the original manuscript.

Nevertheless, for the purpose of completeness the map of AWC can be added to manuscript if referee and editor insist on this topic.

The stripes in Fig. 11 are caused by the process of quantile exchange. In Fig. 2 it is visible that on a small scale the variance of the characteristic is in comparison to the total variance very low. When these values are transferred to their empirical quantile, their numbers differ only from the fifth or sixth digit. A reduction of digits and the application to another empirical distribution function, with values that are again rounded, resulted in the visible stripes.

We are very thankful for this review comment since it pointed out a problem that got lost along the way of preparing this manuscript. We revised and re-applied the resampling scheme, with the results that we were able to remove the stripes (see revised Fig. 11). Changes in the input to sub-basin ascertainment led to slightly different results (rev. Fig. 12) and performances compared to the initial results. However, altered results do not lead to different conclusions.

**Table 1: Normalized reduction of standard deviation for resampled basins**

| Catchment | Pore Volume | | Slope | |
|---|---|---|---|---|
| | $\alpha_1$ [%] | $\alpha_2$ [%] | $\alpha_1$ [%] | $\alpha_2$ [%] |
| Mulde (res) |  58.9 |  26.2 |  8.9 |  76.4 |
| Salzach (res) |  39.7 |  4.8 |  19.7 |  37.5 |

[Figure]

**Figure 1: Resampled AWC values for Mulde and Salzach catchment**

[Figure]

**Figure 12: Results of ACS application for resampled catchments of the Mulde and Salzach (from top to bottom), sub-basins based on resampled pore volume (left) and slope (right). Comparison of $\sigma_U(C)$ and $\sigma_S(C)$ for each application (red and blue lines).**

Changed results have been transferred to the manuscript as on page 13:

Removed line 28/29.

Added following passage in line 30:

"*We also experience a change in performance for slope. The exchange of heights values creates a lower range of slope values and a lower amount of heterogeneity. This patterns resulted in all other applications to inferior $\alpha_2$ performances. Still the geomorphologic structure of the basin remains unchanged and heterogeneity can be assessed by the algorithm (visible through unchanged total reduction).*"

12. "Page 16, line 29, Table 4: Where are the 30 different zones per sub-basin coming from; why so many; how are these zones defined?"

In this section the partition by land cover and heights from Sec. 5.1 is converted into a model structure. As indicated, this separation scheme follows the recommendations by Lindström et al. (1997): Land cover to account for soil and heights for precipitation and evaporation correction factors. Land use has been divided

into forest (all types), bare soil/rock and field (all remaining cells that are not water). Threshold for heights partition has been set to 100m. The obtained spatial setup is shown in Fig. 13 & 14.

Choice of 100m as height threshold is due to the usage in the original HBV publication by Bergström (1976) who used a similar heights subdivision (900m / 10 height-zones) and the HBV model of the German Federal Institute of Hydrology which has been applied in a project at our institute (http://doi.bafg.de/BfG/2016/BfG-1877.pdf).

13. "Page 17, Table 5: As already mentioned in the general comments the comparison of the performance for two model versions with such a large difference in number of parameters needs to be given some more thought. Could the reason for the better model performance not be just because of the smaller number of parameters and therefore the smaller complexity and easier calibration. This might be tested by an additional model version using the same small number of parameters as in the new delineation scheme but applied on the old conventional basin separation (using only 3 zone per sub-basin too)?"

The trade-offs between parameters and model performance are a topic for themselves and we intended to address it just as scantly as possible. Therefore, we tried to stay consistent in our modelling choices. We applied the same basin partition schemes that we applied in the previous sections and used the same parameter coupling scheme. Parameter coupling included only 6 zonal parameters, all remaining zonal parameters were due to calibration. Our decision for consistency resulted in the shown number of parameters for the three models.

On the one hand we agree with your statement that more parameters make the model more complex which could lower the model performance. On the other hand, a higher number of free parameters offers more flexibility to fit the observed behaviour. We expected that a model with more parameters would perform better than the parsimonious setup in the calibration period. Due to the high degree of specialisation (fitted model parameters) we expected a greater loss of accuracy for high parametrised model structures than for the parsimonious. Our results so far show that even in the calibration period the performance of the parsimonious setup is superior. Leading to our conclusion that the advantage of more flexibility, even with the same parameter coupling scheme, does not compensate the value of the information we added to the model. We closed our analysis at this point because we defined our benchmark as the "common" scheme for the HBV$_{96}$ model.

However, as your comment pointed out, we did not consider a benchmark model with a comparable number of parameters but with different spatial resolution. We suggest to stick with the benchmark basin partition scheme and extend the coupling scheme. Previously, parameters that could be somehow reasonably be related to storage information (like SM – soil storage, ICMAX – interception storage etc.) were coupled between the zones within a sub-basin. When we loosen this restriction and take all parameters into the coupling scheme we are able to reduce the number of free parameters (see Tab. RS1). This advanced coupling scheme has been applied to the ACS model as well to gain comparability of calibration effort.

**Table 4: Parameter quantities**

|  | Benchmark *Free* | Benchmark *Coupled* | Benchmark *Advanced* | ACS *Coupled* | ACS *All coupled* |
|---|---|---|---|---|---|
| Sub-basins | 38 | 38 | 38 | 44 | 44 |
| Zones per Sub. | ~30 | ~30 | ~30 | 3 | 3 |
| Parameter (total) | 19562 | 12198 | 1710 | 2244 | 1980 |
| Parameter (per Sub.) | ~495 | ~321 | 45 | 51 | 45 |

Evaluation of the new calibration schemes gives the following results:

**Table 5: Nash-Sutcliffe Efficiencies of Benchmark and ACS-model**

| Simulation (Start-End) | $NSE_{B;Free}$ [-] | $NSE_{B;Coupled}$ [-] | $NSE_{B;Advanced}$ [-] | $NSE_{ACS}$ [-] | $NSE_{ACS;All}$ [-] |
|---|---|---|---|---|---|
| 1995 - 2006 (C) | 0.678 | 0.659 | 0.682 | 0.792 | 0.791 |
| 2007 - 2011 (V) | 0.524 | 0.570 | 0.578 | 0.622 | 0.647 |
| 1984 - 1995 (V) | 0.496 | 0.516 | 0.525 | 0.607 | 0.546 |
| 1951 - 1961 (V) | 0.433 | 0.568 | 0.458 | 0.660 | 0.572 |

[Figure]

**Figure 18: Nash-Sutcliffe Efficiency of ACS- based model and benchmark model, all zonal parameters coupled**

Section of model performance has been extended to include the presented results, Page 16, Line 32:

"*A high number of parameters is believed to be offer a model structure more flexibility to fit the observed data, though its higher complexity might lower its performance. To compare our proposed model structure with a benchmark comprising a similar number of parameters we performed added a third calibration strategy. The performed approach coupled all zonal parameters as described above. This lowered the amount of parameters per sub-basin to 45 for both model setups. As it can be seen in Tab. 4 the total amount of parameters in the benchmark partition is higher than in the new ACS-based partition.*

After the calibration (time period 1995-2006) we evaluated model performance in three validation periods. Two in direct (temporal) neighbourhood to the calibration period and the last at the very beginning of the time series. Model performance has been calculated as the average Nash-Sutcliffe-Efficiency (NSE) (Nash and Sutcliffe, 1970) of all gauges and is tabulated in Tab. 5. Results show that ACS-parametrisations are

superior in all cases. Its increase in performance ranges from 17-52% in comparison to the free- , 11-21% *to the 6-parameter-*coupled benchmark *and 5–19% to the all-coupled parametrisation*.

Beside this "lumped" evaluation we compared the performance of the models at each gauge in each period. Comparison of NSE for *6-parameter coupled* models are shown in Fig. 15, for ACS and free-benchmark model in Fig. 16. *Comparison for the all-coupled parametrisation is shown in Fig. 17*. We can see that the individual performances offer the same conclusion as the lumped performance, though some results are better for benchmark models (both parametrisations). To be more precise, in case of *6-parameter-*coupled models 20 points (rep. a single gauge in one of the time periods) are below equivalency (rep. a better performance of the benchmark model), in case of the free-benchmark model 12 *and for the all-coupled benchmark 23* points. Representing 15%*, 9% and 20% of* evaluated cases."

---

## Author Comment (AC2)

**Response to Reviewer Comment #2**

First, we would like to thank the reviewer for his comments. We will consider your recommendation to, hopefully, overcome the named formal reasons for rejections.

Response to outlined problems:

1. "First problem is that some basic mathematics is unclear, e.g. eqs. (1) and (2). The summation variable in (1) needs clarification. x is running from i·Δs to (i+1)·Δs, but there is no indication of the range of x-values to be applied. In (1) the mean value E(C)i is established, but in the calculation of the standard deviation in (2) the applied mean value is denoted E(C)x"

   We agree on this point. The formulation has been not correctly. We propose the following revision for page 5, after line 25:

   *"The stream flow length is subdivided into multiples of the length Δs (for $x_S$) and the hillslope flow length into multiples of the distances Δo (for $x_H$). In this way defined distance classes split the basin into stripes. Depending on width Δs the basins is classified into $N_S$ distance classes.*
   *Let us consider a single distance class i, where i is an integer indicating the number of the class. All cells of the input data with a flow distance in a range between i·Δs and (i+1) ·Δs are assigned to this distance class. We can write the set of distances of $x_S$ assigned to i as the following set B:*

   $$B = \left[ i \cdot \Delta s; (i+1) \cdot s \right] \tag{1}$$

   *To characterise property C in distance class i we can calculate the expected value E(C) and standard deviation σ(C), taking only values of C into account that lay in a distance the boundaries of the distance classes:*

   $$E(C)_i = \frac{1}{w(i \cdot \Delta s)} \sum_{(j|x_{s;j} \in B)} C_j \quad (2) \qquad \text{and} \qquad \sigma(C)_i = \sqrt{\frac{1}{w(i \cdot \Delta s)} \sum_{(j|x_{s;j} \in B)} \left( C_j - E(C)_i \right)^2} \quad (3)$$

   *Where w(i·Δs) is the number of values (or grid cells) within the class.* Please note that w(x) is the non-normalised value of the area-function (Snell and Sivapalan, 1994)",

   To enhance index consistency we changed the following equation:

   Page 6, line 25, Eq. 3:                 Index j replaced with i, indicating distance class numbers
   Page 7, line 1, Eq. 4 & line 4, Eq. 5   Index j replaced with i, indicating distance class numbers
   Page 9, line 9, Eq. 6                      Index j replaced with b, indicating parallel basins/zones *

   * required further changes in the manuscript after line 10:
   "Where $B_i$ is the number of *parallel basins or zones, $w_b(i \cdot \Delta s)$* the number of cells within the distance-class i of the considered parallel basin/zone and $\sigma(C)_{i;b}$ are the neighbouring standard deviations."

2. "Second problem is the awkward English, which is below an acceptable standard for a high quality journal. There are numerous grammatical errors (punctuation; wrong tense, wrong words, missing words, bad construction of sentences etc.), which is very disturbing for grasping the real content of the paper."

Your comment on linguistic quality and/or style has been stated by the first referee as well. To comply with the standard of the journal the manuscript is under extensive linguistic revision. The revision is not yet completed and will be addressed separately in an additional response

---

## Author Response (AR1)

We would like to thank the editor and all reviewers for their feedback and suggestions that helped to improve the manuscript. All reviewer comments, our responses and the changes made in the manuscript are summarised below. Changes made to the manuscript are shown in blue. References to the position of these changes in manuscript have been updated to the position in the revised manuscript. Tracking of revision is appended to this summary.

**Response to Reviewer Comment #1**

Response to general comment on model comparison is given below under Comment no. 13. Here we will focus on the content-related comments by the referee. Criticism of linguistic quality caused us to repeat and extent our proof-reading procedure (see revision tracking below).

Response to detailed comments:

1. "Page 3, line *20* : The computational demands of semi-distributed models with polygon based delineation is nowadays often negligible."

   We agree that the computational demand to perform a single model run is often negligible, but this omits the demands for model calibration or extensive sensitivity analysis studies. However, the actual formulation does not address this subjects and has been reformulated as follows:

   "The aim of the proposed algorithm is set to offer a basin partition with a minimum of heterogeneity by a minimum of sub-divisions, i.e. to reduce  the number of unnecessary sub-divisions and subsequently the number of parameters in cases of hydrological modelling."
   *[Page 3, Line 30]*

2. "Page 4 and Fig. 2: Pore volume and available water capacity should be defined. For instance is with "total pore volume" the "porosity" meant? Usually these variables are given in volume percent? Here they are given in mm, why? Does this not require an information about the soil depth?"

   Our data basis provides the opportunity to assess soil properties in different depths. In previous projects, where this data has been applied, we experienced that the majority of soil storage is located in the upper zones of the soil layer. Namely a depth of up to 2 meters covers nearly all available soil storage. Why we applied the *[mm]* data and not the *[%]* is in this case due availability (data from previous project) than for any particular consideration.

   The employed TPV data is calculated for a soil depth of up to 2 meters, this information has been added to the manuscript, Legend in Figure 2 and Page 4 Line 18-19 have been revised:

   "To characterise the soil characteristics of the German catchments, a gridded soil data map from the German Federal Institute for Geosciences and Natural Resources (BÜK200) and CORINE land coverage data (CLC) (Bossard et al., 2000) were used. Pedo-transfer functions (Sponagel, 2005) were applied to transfer this information into gridded data about (available) water capacities (AWC), maximum soil

storage capacity (referred as total pore volume TPV) and hydraulic conductivity (HC) for the upper soil, up to two meters depth."

*[Page 4, line 32]*

3. "Page 5, Lines 24-25: I was trying to picture the basin split into stripes. I found Figure 5 as an visual explanation. However, comparing Fig. 5 with Fig. 3 there should be much less points of the distance-factor function in Fig. 3 (only 5 points). Later I realised that Fig. 3 does not belong to Fig. 5. For demonstration purpose I would suggest providing a pair of Figures with distance classes and corresponding distance factor function using a few classes only. May be Fig. 5 needs to come before Fig. 3."

The enhance comprehensibility we follow your recommendation. The distance-factor function of the synthetic catchment has been created (see Figure below) and added to the manuscript as Fig. 4. The synthetic catchment is now introduced on *Page 6, Line 19* as follows:

"Please note that w(x) is the non-normalised value of the area-function (Snell and Sivapalan, 1994). To visualise the proposed function a simple synthetic basin with its stream network, distance-classes and an arbitrary characteristic is shown in Fig. 3. To keep things simple Fig. 3c shows the unified flow length (comprising $x_S$ and $x_H$) derived from the flow direction data in Fig. 3b. As it can be seen in Fig. 3c the basin has been split into 5 distance classes. Applying Eq. 1 and 2 to the data (Fig. 3a) produces the average and standard deviation within these five distance classes. The obtained distance-factor function is shown in Fig. 4.
Figure 5 shows the application to real data in a meso-scale catchment, namely for AWC in the Mulde catchment."

[Figure]

**Figure 3: Distance-factor function of sample Data in synthetic catchment**

4. "Page 5, line 31: (Eq. 1 and 2)"

Already addressed in previous point.

5. "Fig. 4: Use the same units on the x-axis as in Fig. 3."

Figure 4 (now 5) has been revised.

6. "Fig. 5: Example input data are on the "left"!"

   Subscript of Fig. 5 (now 3) has been changed as noted.

7. "Page 12, Eq. 8: I would suggest to explain this equation in words. I also would suggest to exchange the sides of omega and sigma (even if mathematically not necessary) in both numerator and denominator."

   We suppose the suggestion to switch omega and sigma is indented to clarify the summation. Instead of switching the summands we added parenthesis for clarification. An explanation of Eq. 9 has been added to the manuscript on *page 13, line 4:*

   "In addition to this we applied a second performance metric to evaluate to what extend our target has been met. The metric $\alpha_2$ highlights cases where the total heterogeneity was decreased significantly, however, but still being above objective as defined by threshold $\Omega$. The metric $\alpha_2$ was calculated as the delta between threshold $\Omega$ and standard deviation in the separated catchment $\sigma_S(C)$, normalised by the delta between $\Omega$ and the standard deviation in the unseparated catchment $\sigma_U(C)$:

   $$\alpha_2 = \frac{\sum_{i \in M(S)} \left( \Omega - \sigma_{S;i}(C) \right)}{\sum_{j \in M(U)} \left( \Omega - \sigma_{U;j}(C) \right)} \tag{9}"$$

8. "Page 12, line 15: What are cases with "negative outcome"? Do you mean insufficient variance reduction?"

   That is correct. Page 13, line 21 has been revised as suggested:

   "Focussing on cases with insufficient variance reduction we were able to identify some limitations for the algorithm"

9. Fig. 8, 9 and 12: The distance factor functions are hard to read. Use vector graphics and/or large fonts and/or larger figures.

   Figures have been revised. Figures are now embedded in EPS format, fonts have been enlarged. Figures will be submitted separately for better processing.

10. "Fig. 10: In the legend of AWC "max" and "min" need to be exchanged."

    Legend of Fig. 11 has been revised.

11. Fig. 11: When resampling of AWC for the Mulde river basin is shown also the original AWC map of the Mulde basin should be shown for comparisons. Why are there some kind of horizontal stripes in Fig. 11?

The map of original AWC values in the Mulde basin is not shown in the manuscript, because the spatial arrangement of AWC and the shown TPV is identical (as mentioned on *page 14, line 24*) due to origin of these data grids. Both values follow the arrangement of the soil map, their only difference is the range of values (also given in *line 24*), caused by different calculation formula (Sponagel, 2005). Since the manuscript already has a great number of figures and the information content of the AWC figure low we omitted it in the original manuscript.

The stripes in Fig. 11 (now 12) are caused by the process of quantile exchange. In Fig. 2 it is visible that on a small scale the variance of the characteristic is in comparison to the total variance very low. When these values are transferred to their empirical quantile, their numbers differ only from the fifth or sixth digit. A reduction of digits and the application to another empirical distribution function, with values that are again rounded, resulted in the visible stripes.

We are very thankful for this review comment since it pointed out a problem that got lost along the way of preparing this manuscript. We revised and re-applied the resampling scheme, with the results that we were able to remove the stripes (see revised Fig. 12). Changes in the input to sub-basin ascertainment led to slightly different results (rev. Fig. 13) and performances compared to the initial results. However, altered results do not lead to different conclusions.

**Table 1: Normalized reduction of standard deviation for resampled basins**

| Catchment | Pore Volume | | Slope | |
|---|---|---|---|---|
| | $\alpha_1$ [%] | $\alpha_2$ [%] | $\alpha_1$ [%] | $\alpha_2$ [%] |
| Mulde (res) | 58.9 | 26.2 | 8.9 | 76.4 |
| Salzach (res) | 39.7 | 4.8 | 19.7 | 37.5 |

[Figure]

**Figure 12: Resampled AWC values for Mulde and Salzach catchment**

[Figure]

**Figure 13: Results of ACS application for resampled catchments of the Mulde and Salzach (from top to bottom), sub-basins based on resampled pore volume (left) and slope (right). Comparison of $\sigma_U(C)$ and $\sigma_S(C)$ for each application (red and blue lines).**

Changed results have been transferred to the manuscript as on *page 15, line 13*:

"We also experienced a change in performance for the slope. The exchange of heights values led to a lower range of slope values and a lower amount of heterogeneity. These patterns resulted in all other applications in inferior $\alpha_2$ performances. Still, the geomorphologic structure of the basin remained unchanged and heterogeneity could be assessed by the algorithm (visible through unchanged total reduction)."

12. "Page 16, line 29, Table 4: Where are the 30 different zones per sub-basin coming from; why so many; how are these zones defined?"

In this section the partition by land cover and heights from Sec. 5.1 is converted into a model structure. As indicated, this separation scheme follows the recommendations by Lindström et al. (1997): Land cover to account for soil and heights for precipitation and evaporation correction factors. Land use has been divided into forest (all types), bare soil/rock and field (all remaining cells that are not water). Threshold for heights partition has been set to 100m. The obtained spatial setup is shown in Fig. 13 & 14.

Choice of 100m as height threshold is due to the usage in the original HBV publication by Bergström (1976) who used a similar heights subdivision (900m / 10 height-zones) and the HBV model of the German Federal Institute of Hydrology which has been applied in a project at our institute (http://doi.bafg.de/BfG/2016/BfG-1877.pdf).

13. "Page 17, Table 5: As already mentioned in the general comments the comparison of the performance for two model versions with such a large difference in number of parameters needs to be given some more thought. Could the reason for the better model performance not be just because of the smaller number of parameters and therefore the smaller complexity and easier calibration. This might be tested by an additional model version using the same small number of parameters as in the new delineation scheme but applied on the old conventional basin separation (using only 3 zone per sub-basin too)?"

The trade-offs between parameters and model performance are a topic for themselves and we intended to address it just as scantly as possible. Therefore, we tried to stay consistent in our modelling choices. We applied the same basin partition schemes that we applied in the previous sections and used the same parameter coupling scheme. Parameter coupling included only 6 zonal parameters, all remaining zonal parameters were due to calibration. Our decision for consistency resulted in the shown number of parameters for the three models.

On the one hand we agree with your statement that more parameters make the model more complex which could lower the model performance. On the other hand, a higher number of free parameters offers more flexibility to fit the observed behaviour. We expected that a model with more parameters would perform better than the parsimonious setup in the calibration period. Due to the high degree of specialisation (fitted model parameters) we expected a greater loss of accuracy for high parametrised model structures than for the parsimonious. Our results so far show that even in the calibration period the performance of the parsimonious setup is superior. Leading to our conclusion that the advantage of more flexibility, even with the same parameter coupling scheme, does not compensate the value of the information we added to the model. We closed our analysis at this point because we defined our benchmark as the "common" scheme for the $HBV_{96}$ model.

However, as your comment pointed out, we did not consider a benchmark model with a comparable number of parameters but with different spatial resolution. We suggest to stick with the benchmark basin partition scheme and extend the coupling scheme. Previously, parameters that could be somehow reasonably be related to storage information (like SM – soil storage, ICMAX – interception storage etc.) were coupled between the zones within a sub-basin. When we loosen this restriction and take all parameters into the coupling scheme we are able to reduce the number of free parameters (see Tab. RS1). This advanced coupling scheme has been applied to the ACS model as well to gain comparability of calibration effort.

**Table 4: Parameter quantities**

|                    | Benchmark *Free* | Benchmark *6-Coupled* | Benchmark *All coupled* | ACS *6-Coupled* | ACS *All coupled* |
|--------------------|-----------------|----------------------|-------------------------|-----------------|-------------------|
| Sub-basins         | 38              | 38                   | 38                      | 44              | 44                |
| Zones per Sub.     | ~30             | ~30                  | ~30                     | 3               | 3                 |
| Parameter (total)  | 19562           | 12198                | 1710                    | 2244            | 1980              |
| Parameter (per Sub.) | ~495          | ~321                 | 45                      | 51              | 45                |

Evaluation of the new calibration schemes gives the following results:

**Table 5: Nash-Sutcliffe Efficiencies of Benchmark and ACS-model**

| Simulation (Start-End) | $NSE_{B;Free}$ [-] | $NSE_{B;6}$ [-] | $NSE_{B;All}$ [-] | $NSE_{ACS;6}$ [-] | $NSE_{ACS;All}$ [-] |
|------------------------|------------------|----------------|-------------------|-------------------|---------------------|
| 1995 - 2006 (C)        | 0.678            | 0.659          | 0.682             | 0.792             | 0.791               |
| 2007 - 2011 (V)        | 0.524            | 0.570          | 0.578             | 0.622             | 0.647               |
| 1984 - 1995 (V)        | 0.496            | 0.516          | 0.525             | 0.607             | 0.546               |
| 1951 - 1961 (V)        | 0.433            | 0.568          | 0.458             | 0.660             | 0.572               |

[Figure]

**Figure 18: Nash-Sutcliffe Efficiency of ACS- based model and benchmark model, all zonal parameters coupled**

Section of model performance has been extended to include the presented results, *Page 18, Line 15*:

"A high number of parameters is assumed to offer a model structure with a higher flexibility to match the observed data, though its higher complexity might lower its performance. To compare our proposed model structure with a benchmark at a similar number of parameters we added a third calibration strategy. The performed approach coupled all zonal parameters as described above. This lowered the amount of parameters per sub-basin to 45 for both model setups. As it can be seen in Tab. 4 the total amount of parameters in the benchmark partition is higher than in the new ACS-based partition.

After the calibration (time period 1995-2006) we evaluated the model's performance in three validation periods. Two in direct (temporal) neighbourhood to the calibration period and the last at the very beginning of the time series. Model performance has been calculated as the average Nash-Sutcliffe-Efficiency (NSE) (Nash and Sutcliffe, 1970) of all gauges and is tabulated in Tab. 5. Results show that

ACS-parametrisations are superior in all cases. Its increase in performance ranges from 17-52% in comparison to the free-, 11-21% to the 6-parameter-coupled benchmark and 5-19% to the all-coupled parametrisation.

Beside this "lumped" evaluation we compared the performance of the models at each gauge in each period. A comparison of NSE for 6-parameter coupled model is shown in Fig. 16, for ACS and free-benchmark model in Fig. 17. Comparison for the all-coupled parametrisation is shown in Fig. 18. We can see that the individual performances led to the same conclusion as the lumped performance, though some results are better for benchmark models (both parametrisations). To be more precise, in case of 6-parameter coupled model 20 points (rep. a single gauge in one of the time periods) are below equivalency (rep. a better performance of the benchmark model), in case of the free-benchmark model 12 and for the all-coupled benchmark 23 points. Representing 15%, 9% and 20% of the evaluated cases.

**Response to Reviewer Comment #2**

Response to outlined problems:

1. "First problem is that some basic mathematics is unclear, e.g. eqs. (1) and (2). The summation variable in (1) needs clarification. x is running from $i \cdot \Delta s$ to $(i+1) \cdot \Delta s$, but there is no indication of the range of x-values to be applied. In (1) the mean value $E(C)_i$ is established, but in the calculation of the standard deviation in (2) the applied mean value is denoted $E(C)_x$"

We agree on this point. The formulation has been not correctly. We propose the following revision for page 6, after line 5:

"The stream flow length was subdivided into multiples of the length $\Delta s$ (for $x_S$) and the hillslope flow length subdivided into multiples of the distances $\Delta o$ (for $x_H$). Distance classes defined in this way will split the basin into stripes. Depending on the width $\Delta s$ the basin would be classified into $N_S$ distance classes, where $N_S$ is an integer larger or equal to one.

Let us now look at a single distance class $i$, where $i$ indicates the class. All cells of the input data with a flow distance in a range between $i \cdot \Delta s$ and $(i+1) \cdot \Delta s$ are assigned to this distance class. We can write the set of distances of $x_S$ assigned to $i$ as the following set $B$:

$$B = \left[ i \cdot \Delta s ; (i+1) \cdot \Delta s \right] \tag{1}$$

To characterise a property C in the distance class $i$ we can calculate the expected value $E(C)$ and standard deviation $\sigma(C)$, taking only those values of C into account that are situated in the boundaries of the distance classes:

$$E(C)_i = \frac{1}{w(i \cdot \Delta s)} \sum_{(j | x_{s;j} \in B)} C_j \quad (2) \qquad \text{and} \qquad \sigma(C)_i = \sqrt{\frac{1}{w(i \cdot \Delta s)} \sum_{(j | x_{s;j} \in B)} \left( C_j - E(C)_i \right)^2} \quad (3)$$

$w(i \cdot \Delta s)$ represents the number of values (or grid cells) within the class. Please note that w(x) is the non-normalised value of the area-function (Snell and Sivapalan, 1994).",

To enhance index consistency we changed the following equation:

Page 6, line 15, Eq. 3:                     Index j replaced with i, indicating distance class numbers

Page 7, line 20, Eq. 4 & line 23, Eq. 5     Index j replaced with i, indicating distance class numbers

Page 10, line 9, Eq. 7                       Index j replaced with b, indicating parallel basins/zones *

\* required further changes in the manuscript after line 10:

"with $P_i$ being the number of parallel basins or zones, $w(i{\cdot}\Delta s)_p$ the number of cells within the distance-class $i$ of the considered parallel basin/zone $b$ and $\sigma(C)_{i;p}$ being the neighbouring standard deviations"

2. "Second problem is the awkward English, which is below an acceptable standard for a high quality journal. There are numerous grammatical errors (punctuation; wrong tense, wrong words, missing words, bad construction of sentences etc.), which is very disturbing for grasping the real content of the paper."

Your comment on linguistic quality and/or style has been stated by the first referee as well. To comply with the standard of the journal the manuscript is underwent extensive linguistic revision. Again, the result of linguistic revision can be seen below.

**Response to Reviewer Comment #3**

Your first point has been stated by all reviewers so far and we truly accept this critique. Results of this revision, again, can be tracked below.

The second point intersects with the readability. You suggest a more technical, mathematical description of the algorithm. In the former submission of this manuscript we focused on a technical description of the algorithm and its functionality (but based on the reviews we decided to withdraw the manuscript and focus more on modelling application). A major point of the revisions for the previous manuscript was a too technical depiction of the algorithm that did not match the requirements of a research paper. Hence, we omitted the detailed description and offered a "verbose" qualitative description. We added the appendix to show the functionality of the involved tools. In its actual form, the manuscript is intended to offer a trade-off between technical detail and facile qualitative description.

Nevertheless, the sequence of the algorithm is shown and described in the manuscript. For readers with more interest in the algorithm and its implementation we made the code available as supplementary to this paper.

Besides the structure of the paper, you pointed out that the background of the threshold $\Omega$ and the non-linearity coefficient $e$ is not made clear.

As described in Sect. 3.2.1 the threshold $\Omega$ is used to distinguish between distance classes comprising "high" and "low" variance. Since we do not know the exact amount of tolerable variance we had to come up with a concept to value our target. There are several ways to do so. One way might be to just take an arbitrary percentile of present variance, a valuation by cluster analysis or by setting it to a fixed number.

But the threshold, and hence the number of ascertained sub-basins, is dependent on the purpose of the performed partition. For some applications a detailed subdivision, bringing the heterogeneity to a minimum (will result in

more computational effort), might be desirable while for other purposes a facile partition might be sufficient. We intended to implement a flexible function that unites the requirements for both purposes (and in between).

The weighting factor $\omega$ offers this kind of flexibility and $e$ is its parameter. If $e$ tends to zero, the threshold $\Omega$ tends to the average of present standard deviation. The factor $e$ set to this value should be used in applications were only regions with significantly higher heterogeneity are to be captured. If $e$ increases the threshold lowers and hence the number of subdivisions is likely to increase. With an increasing factor $e$ more heterogeneity is captured by the algorithm.

Though it is interesting which value for $e$ is suitable for what kind of application we omitted deliberately a detailed analysis. In the present state of the manuscript $e$ is a user-dependent specification and its introduction aims to lower the range of possible arbitrary threshold values. In the following figure Fig. RC1 we compared the value of $\Omega$ to percentiles of present standard deviation values in the Mulde catchment. As it can be seen we approximately bisect the range of possible values. All figures in the manuscript involving $\Omega$ use three different values of $e$, namely 0., 0.5 and 1. Our intention was to show a possible range of $\Omega$ in our application, though we constantly set $e = 0.5$. All this is indeed missing in the manuscript because we ascribed only minor importance on this factor.

[Figure]

**Figure RC1: Percentile of $\Omega$ against non-linearity factor $e$**

To make the concept of the non-linearity factor $e$, its chosen value in the application and its constant depiction in the distance-factor functions understandable we added the following lines on page 8, line 4:

"As the index $N_S$ in Eq. (3) indicates, the threshold $\Omega$ is calculated for the entire catchment and does not necessarily represent the true value of $\sigma(C)$ for homogeneous sub-basins. It just helps to distinguish regions of high and low variances. Therefore it is recommended to vary the value of the non-linearity factor $e$ in a range of $[0; \infty]$, though values between 0 and 1 were found most applicable in our case studies. If $e$ is set to zero $\Omega$ is equal to the average of $\sigma(C)$ in the basin. As $e$ is increased $\Omega$ lowers. The choice of $e$ is dependent on the intention of the purpose of partition. If a detailed subdivision is required to capture the majority of heterogeneity $e$ should be set to value greater or equal to 1. Otherwise if only regions with a significantly higher heterogeneity are to be captured $e$ is recommended to be set to 0. However, in the presented case studies $e$ has been set to 0.5. To indicate the range of potential results $\Omega$ is shown in all distance-factor functions for values of e ~ 0., 0.5 and 1.0."

To your last sub-point of your critique on the depiction of the algorithm we would like to give a detailed response. The revision of mathematical equations has been pursued from the response to Comment #2 as follows:

We introduced a set $B$ of stream-flow distances $x_S$ to the description of the distance-factor function (Page 6, Line 12):

$$B = \left[ i \cdot \Delta s ; (i+1) \cdot \Delta s \right] \tag{1}$$

The variable $i$ is an integer ranging from 1 to $N_S$, the number of distance classes in the basin.

This set is used afterwards in the equations for $E(C)$ and $\sigma(C)$:

$$E(C)_i = \frac{1}{w(i \cdot \Delta s)} \sum_{(j | x_{s;j} \in B)} C_j \tag{2}$$

$$\sigma(C)_i = \sqrt{ \frac{1}{w(i \cdot \Delta s)} \sum_{(j | x_{s;j} \in B)} \left( C_j - E(C)_i \right)^2 } \tag{3}$$

Equation 3 (now 4), [*Page 7, line 20*] the beforehand discussed threshold $\Omega$, had also not been compiled correctly. As you pointed out the factor $e$ has to appear in the nominator and denominator:

$$\Omega = \frac{\sum_{i=1}^{N_S} \omega_i^e \cdot \sigma(C)_i}{\sum_{i=1}^{N_S} \omega_i^e} \tag{4}$$

Furthermore, the weighting factor Eq. 4 (now 5) on *page 7, line 23*, has been changed to make the range of Min/Max Function clear:

$$\omega_i = \frac{\sigma(C)_i - \max_{1 \leq i \leq N_S} \left( \sigma(C)_i \right)}{\min_{1 \leq i \leq N_S} \left( \sigma(C)_i \right) - \max_{1 \leq i \leq N_S} \left( \sigma(C)_i \right)} \tag{5}$$

The objective function Eq. 5 (now 6) has been updated analoguesly [*Page 7, line 26*]:

$$Z = \left\| \left[ i \mid 0 \leq i \leq N_S ; \sigma(C)_i > \Omega \right] \right\| \rightarrow \min \tag{6}$$

On page 9, line 9, Eq.6 (now 7) the evaluation scheme has been updated as well. We added a new index $p$ to denote parallel sub-basins (We changed the index character to prevent a mistake with the set B (Eq. 1)). Index $p$ is an integer ranging from 1 to $P_i$, the number of parallel sub-basins within the original basin $i$ ( $i \in \left[ 0 ; N_S \right]$ ). Remaining indices have been matched with the general notations:

$$\sigma_S(C)_i = \frac{1}{\sum_{p=1}^{P_i} w(i \cdot \Delta s)_p} \cdot \sum_{p=1}^{P_i} \sigma(C)_{i;p} \cdot w(i \cdot \Delta s)_p \tag{7}$$

Your last comment addresses the model application. You suggest to substitute the NSE-based calibration study with a parameter uncertainty analysis or to omit the model application.

Likewise to your critique of the manuscript structure and algorithm depiction we shared your opinion. Moreover the previous submission of the paper changed our attitude. While the detailed technical description was heavily criticised in the previous manuscript, the missing model application "where a model with the new subdivision

outperforms another model" was strongly recommended. So (again) this chapter has been incorporated to meet such expectations. A reduction to the variance topic would be contrary to the aims of this manuscript, defined in the introduction (and title), that clearly sets for modelling purposes. Since we expected that many readers would ask for that missing modelling aspect we would like to keep this section in the manuscript.

Nevertheless, our main intention is to present the method, its ability to capture catchment heterogeneity and the insights on catchment organisation. This is why we only performed a short and simple modelling study with 3 (now 5, see Response to reviewer comment #1) calibrated model setups.

The problem with NSE-based comparisons is indeed their dependence on the applied calibration procedure. Therefore, we tried to apply the same calibration strategy to both spatial setups, titled "coupled" calibration for ACS and the benchmark setup. In both applications the same 6 parameters are coupled and the identical calibration tool and the same number of iterations has been used. The only thing changed is the coupling-information, the land cover in the benchmark setup and TPV-values in the ACS setup. This procedure is described on *page 17 from line 27 to page 18, line 9*.

Additionally, we allowed the benchmark setup to exploit its full number of parameters in an unconstraint calibration run, involving the same calibration tool and number of iterations (as described from line 7 to 9). As reviewer #1 pointed out we had a parameter mismatch in this calibration study and we therefore introduced a second parameter coupling applied to all zonal-parameters. The results and changes in manuscript can be found in our response to the first reviewer.

Your suggestion on parameter uncertainty seems quite appealing, though. Therefore, we performed a Monte-Carlo Simulation with 10'000 random parameterisations and saved the resulting NSE and parameters for several sub-basin in the Mulde catchment. The Monte-Carlo simulation has been applied to the all-parameter coupling scheme. We evaluated our results scantly with a GLUE approach. We used the NSE as generalised likelihood function and set a threshold of 1% for behavioural parameter sets. Parameter uncertainty has been calculated with the following equation:

$$U(Par) = \sqrt{\frac{\sum_{x \in Nx} \left( Par_x - \bar{P} \right)^2 \cdot NSE_x}{\sum_{x \in Nx} NSE_x}}$$

where *Par* is the analysed parameter, *Nx* is the behavioural parameter set, *NSE_x* is the generalised likelihood of the parameter set and $\bar{P}$ the average parameter in set *Nx*:

$$\bar{P} = \frac{1}{|Nx|} \cdot \sum_{x \in Nx} Par_x$$

To compare the uncertainty of all parameters their actual values have been normalised by their parameter constraints. In Tab. RC2 all parameters are given with their lower and upper boundaries, grouped by their usage in the model. "Single" parameters are used in the ground water levels and post-processing of the model, "Coupled" parameters are used in each individual zone. Since we applied the all-parameter coupling scheme, these parameters are estimated once per sub-basin and by a linear coupling coefficient transferred to all zones (See response to Comment #1). The coupling parameters are not shown in Tab. RC2 but their values are uniformly in the interval [0, 2].

**Table RC2: Parameter constraints HBV₉₆ model**

| Parameter | Lower Boundary | Upper Boundary |
|---|---|---|
| **Single** | | |
| ALPHA | 0 | 2 |
| K4 | 0.001 | 0.5 |
| KHQ | 0.01 | 0.5 |
| PERCMAX | 0 | 5 |
| MAXBAZ | 0 | 1 |
| **Coupled** | | |
| BETA | 1 | 4 |
| CFLUX | 0 | 2 |
| CFMAX | 1 | 6 |
| CFR | 0 | 2 |
| DTTM | -2 | 2 |
| ECORR | 0.95 | 1.05 |
| EPF | 0 | 0.2 |
| ERED | 0 | 0.5 |
| ETF | 0 | 0.2 |
| FC | 100 | 500 |
| ICMAX | 1 | 20 |
| LP | 0.75 | 1 |
| RFCF | 0.95 | 1.05 |
| SFCF | 0.95 | 1.05 |
| TT | -2 | 2 |
| TTINT | 1 | 5 |
| WHC | 0 | 2 |

Following figures RC3-6 we plot $\bar{P}$ in the normalised parameter space for ACS setup and the benchmark setup for a single sub-basin, where the shown error bars represent calculated *U(Par)* values.

[Figure]

**Figure RC3: Single parameters**

[Figure]

**Figure RC4: Coupled parameters**

[Figure]

**Figure RC5: Coupling parameters**

As it can be seen in the figures above the uncertainty indicators are symmetrical for the most parameters, meaning that the uncertainty is equal for both models. Uncertainty values for the benchmark and ACS setup plotted against each other are given in the following Fig. RC6:

[Figure]

In Fig. RC6 several points are located above the "Equality" line, meaning that parameter uncertainty for an individual parameter is higher for the benchmark model than for ACS-model. But nearly the same number is below the line.

These results indicate that the parameter uncertainty could not be lowered (significantly). But this is only a fast and simple evaluation and its result might change with an elaborate technique.

Considering the length of the manuscript we preferred not to include these results (or similar) in the paper, because its explanation (method, model, parameters, etc.) would effort a lot of additional information that are irrelevant for the ACS algorithm.